# Safe Time-Varying Optimization based on Gaussian Processes with Spatio-Temporal Kernel

**Jialin Li**
ETH Zürich (currently with UIUC)
`lijial@ethz.ch`

**Marta Zagorowska**
NTNU (currently with TU Delft)
`m.a.zagorowska@tudelft.nl`

**Giulia De Pasquale**
Eindhoven Univeristy of Technology
`g.de.pasquale@tue.nl`

**Alisa Rupenyan**
Zürich University of Applied Sciences
`alisa.rupenyan@zhaw.ch`

**John Lygeros**
ETH Zürich
`jlygeros@ethz.ch`

## Abstract

Ensuring safety is a key aspect in sequential decision making problems, such as robotics or process control. The complexity of the underlying systems often makes finding the optimal decision challenging, especially when the safety-critical system is time-varying. Overcoming the problem of optimizing an unknown time-varying reward subject to unknown time-varying safety constraints, we propose TVSAFEOPT, a new algorithm built on Bayesian optimization with a spatio-temporal kernel. The algorithm is capable of safely tracking a time-varying safe region without the need for explicit change detection. Optimality guarantees are also provided for the algorithm when the optimization problem becomes stationary. We show that TVSAFEOPT compares favorably against SAFEOPT on synthetic data, both regarding safety and optimality. Evaluation on a realistic case study with gas compressors confirms that TVSAFEOPT ensures safety when solving time-varying optimization problems with unknown reward and safety functions.

## 1  Introduction

We seek to interactively optimize an unknown time-varying reward function $f : \mathcal{X} \times \mathcal{T} \to \mathbb{R}$, where $\mathcal{X}$ is a finite set of decisions, and $\mathcal{T} := \{0, 1, 2, \ldots, T\}$, $T \in \mathbb{N}_+$ denotes the discretized time domain. We assume that the optimization problem is safety-critical, that is, there are constraints that evaluated decisions must satisfy with high probability. Similar to the reward, the constraints are also unknown and potentially time-varying, encoded through $c_i : \mathcal{X} \times \mathcal{T} \to \mathbb{R}$, $i \in \mathcal{I}_c := \{1, 2, \ldots, m\}$, where $m \in \mathbb{N}_+$ denotes the number of safety constraints. The optimization problem at time $t$ is

$$
\begin{aligned}
\max_{\mathbf{x} \in \mathcal{X}} \ & f(\mathbf{x}, t) \\
\text{subject to } & c_i(\mathbf{x}, t) \geq 0, \ i \in \mathcal{I}_c
\end{aligned}
\tag{1}
$$

Both the reward function and the safety constraints are assumed to be unknown but can be evaluated. This is a plausible setting, for example, for UAV that need to perform rescue missions in dangerous and poorly lit environments.

38th Conference on Neural Information Processing Systems (NeurIPS 2024).

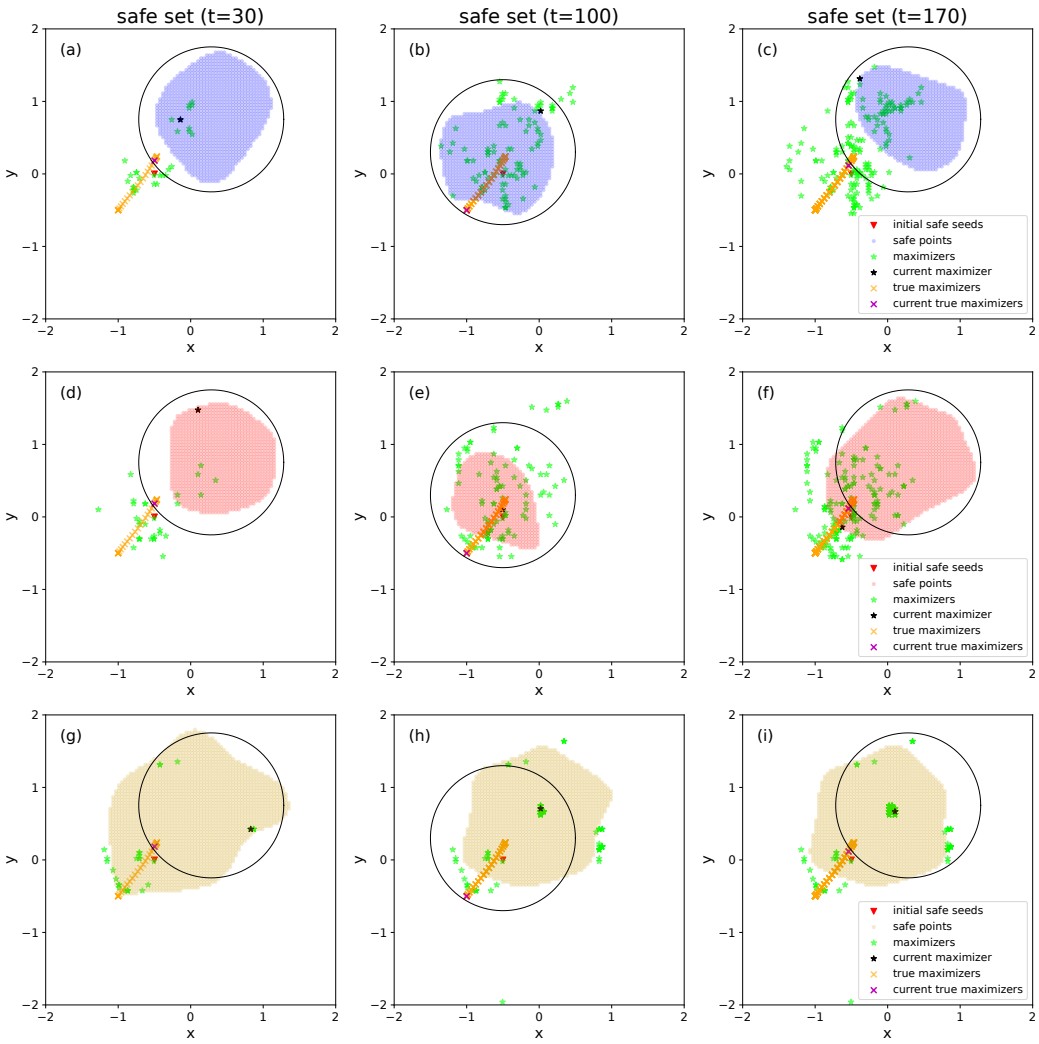

Figure 1: Comparison of safe sets computed by TVSAFEOPT (top row), ETSAFEOPT (middle row), and SAFEOPT (bottom row) at $t = 30$, $t = 100$, and $t = 170$. Because TVSAFEOPT takes the possible changes in time into consideration, the safe sets computed by TVSAFEOPT are contained in the ground truth safe regions while those computed by ETSAFEOPT and SAFEOPT have multiple violations. The reason for the violations in ETSAFEOPT is that the algorithm is unable to detect small changes in the constraints, confirming that the performance of ETSAFEOPT depends on the event detection algorithm.

## 1.1 Related Work

Bayesian Optimization (BO) is a well-established approach for interactively optimizing unknown reward functions. Various BO based approaches have been proposed to solve a wide range of problems in robotics [1, 2], combinatorial optimization [3], sensor networks [4], and automatic machine learning [5, 6]. However, Safe Bayesian Optimization in the time-varying setting is still under-explored.

**Safe Bayesian Optimization** To address safety requirements in safety-critical applications, Safe Bayesian Optimization (SBO) [7] has been proposed to avoid unsafe decisions with high probability by interactively optimizing a reward function under safety constraints. SAFEOPT [7], one of the first SBO algorithms, expands an initial safe set iteratively based on new evaluations and an

Table 1: Overview of safe learning methods based on BO for time-varying problems.

| | Handling Changes in Time | Safety Guarantee | Optimality Guarantee | Safe Seed |
|---|---|---|---|---|
| A-GOOSE [21, 17] | Spatio-temporal kernel | ✓ | ✗ | For all $t$ |
| C-SAFEOPT [8] | Spatio-temporal kernel | ✓ | ✗ | For all $t$ |
| ETSAFEOPT [22] | Event detection | ✗ | ✗ | For all $t$ |
| TVSAFEOPT (ours) | Spatio-temporal kernel | ✓ | ✓ | For initial $t$ |

updated Gaussian Process (GP) model of safety functions. It calculates two subsets, maximizers and expanders, from the current safe set and selects the most uncertain decision within their union to balance maximizing the reward function and expanding the safe set. Subsequent algorithms extend SAFEOPT to handle multiple constraints [8], decouple safe set expansion from optimization [9], and expand the safe set in a goal-oriented manner [10]. These methods also explore disconnected safe regions [11, 12] and enhance information-theoretic efficiency [13, 14]. They have been applied to controller tuning for a ball-screw drive [15] and quadrupeds [16], and adaptive control on a rotational motion system [17]. However, SBO typically does not take into account changes with time.

**Contextual Bayesian Optimization** Contextual Bayesian Optimization (CBO) has been introduced to address the influence of external environmental factors on reward and safety functions. Krause and Ong [18] extend the Gaussian Process Upper Confidence Bound (GP-UCB) algorithm [19] by incorporating contextual variables into unconstrained BO, demonstrating sub-linear regret analogous to GP-UCB. An advancement of this framework is proposed in [20], with the Safe Contextual GP-UCB optimizing the contextual upper confidence bound within a safe set to manage room temperature via a PID controller. Berkenkamp et al. [8] present a contextual adaptation of SAFEOPT, discussing its safety and optimality guarantees by framing contextual SBO as distinct SBO sub-problems. Additionally, König et al. [21] extends GOOSE [10] to the contextual domain for model-free adaptive control scenarios. Similarly to SBO, CBO does not explicitly consider time-varying problems.

**Time-Varying Bayesian Optimization** Time-Varying Bayesian Optimization (TVBO) addresses problems where the objective is time-dependent, modeled with a temporal kernel [23]. Methods in this setting include periodical resetting [23], change detection [24, 25], sliding-window approaches using recent data [26], and discounting via exponentially decaying past observations [27]. However, these techniques have been developed for unconstrained BO and are unsuitable for safety-critical applications.

**Time-Varying Safe Bayesian Optimization** In the safety-critical time-varying setting, contextual lower confidence bounds can be optimized within the safe set [20], but it does not guarantee optimality theoretically. An event triggering mechanism is introduced to SBO to restart exploration from a backup policy [22], but it may not trigger reliably during changes, posing a safety risk. Extensions to SBO with contextual variables provide theoretical safety and optimality analyses [8, 16], treating contextual SBO as separate sub-problems for each contextual value, and assuming an initial safe set for each. However, ensuring optimality requires each contextual value to appear frequently, which is impractical in time-varying scenarios.

## 1.2 Methodology and Contributions

**Methodology** We propose the TVSAFEOPT algorithm to optimize an unknown time-varying reward subject to unknown time-varying safety constraints. The algorithm focuses on Time-Varying Safe Bayesian Optimization (TVSBO). TVSAFEOPT utilizes a spatio-temporal kernel and time Lipschitz constants as prior knowledge about how the problem depends on time. The temporal part of the kernel encodes the continuity of the functions with time while the Lipschitz constants explicitly provide upper bounds on how fast the functions may change. Instead of considering safe sets at previous iteration as safe at the current iteration, which might lead to unsafe decisions, TVSAFEOPT robustly subtracts the safety margin when updating the safe sets (Figure 1). In this way, the algorithm is capable of adapting in real time and guarantees safety even when exploring the safe region of non-stationary problems.

**Contributions** Our contributions are threefold: a) We propose the TVSAFEOPT algorithm based on Gaussian processes with spatio-temporal kernels; b) We provide formal safety guarantees for TVSAFEOPT in the most general time-varying setting and optimality guarantees for TVSAFEOPT for

locally stationary optimization problems; c) We show TVSAFEOPT performs well in the most general time-varying setting both on synthetic data and on a realistic case study on gas compressors. In Table 1, we compare Adaptive GOOSE, Contextual SAFEOPT, ETSAFEOPT, and TVSAFEOPT in terms of how they handle changes in time, safety guarantees, optimality guarantees, and required safe seeds.

### 1.2.1 Expected societal impact

The TVSAFEOPT algorithm proposed in this paper extends the state of the art in Time-Varying Safe Bayesian Optimization by enabling solving optimization problems with time-varying reward and constraints without pre-defining the time changes that can be compensated.Thus, the algorithm can be used at the design stage of operating strategies for safety-critical systems, such as medical dosage design [28] and controller design in robotics [17], or during online operation of chemical plants [29] or autonomous racing [30].

## 2 TVSAFEOPT Algorithm

The TVSAFEOPT algorithm builds upon SAFEOPT [7], to handle time-varying reward function and safety functions. The key new feature of TVSAFEOPT is its capability of safely transferring the current safe set to the next time step. TVSAFEOPT achieves this with the help of the spatio-temporal kernel as well as the sequence of time Lipschitz constants. The approach is summarized in Algorithm 1.

### 2.1 Assumptions

Following [8], we incorporate the reward and safety functions into an auxiliary function $h : \mathcal{X} \times \mathcal{T} \times \mathcal{I} \to \mathbb{R}$, where $\mathcal{I} := \{0\} \cup \mathcal{I}_c$,

$$h(\mathbf{x}, t, i) := \begin{cases} f(\mathbf{x}, t) & \text{, if } i = 0 \\ c_i(\mathbf{x}, t) & \text{, if } i \in \mathcal{I}_c \end{cases} \tag{2}$$

We model the auxiliary function using a prior Gaussian Process (GP) with zero mean and spatio-temporal kernel $\kappa : (\mathcal{X} \times \mathcal{T} \times \mathcal{I}) \times (\mathcal{X} \times \mathcal{T} \times \mathcal{I}) \to \mathbb{R}$, [31]. We require $h$ to be Lipschitz continuous with respect to both $\mathbf{x}$ and $t$, and to have bounded norm in the Reproducing Kernel Hilbert Space (RKHS) [32] associated with the kernel $\kappa$ as formalized in the following.

**Assumption 2.1.** The spatio-temporal kernel is positive definite, and satisfies $\kappa((\mathbf{x}, t, i), (\mathbf{x}, t, i)) \leq 1$, for all $\mathbf{x} \in \mathcal{X}, t \in \mathcal{T}, i \in \mathcal{I}$. The function $h(\mathbf{x}, t, i)$ has bounded norm in the RKHS associated with kernel $\kappa$. The function $h(\mathbf{x}, t, i)$ is $L_{\mathbf{x}}$-Lipschitz continuous with respect to $\mathbf{x}$ in the domain $\mathcal{X}$ with respect to some metric $d : \mathcal{X} \times \mathcal{X} \to \mathbb{R}_{\geq 0}$ for all $t \in \mathbb{N}$, $i \in \mathcal{I}$. There exists a sequence $\{L(t)\}_{t \in \mathbb{N}, t < T}$, such that, for all $\mathbf{x} \in \mathcal{X}$, $i \in \mathcal{I}$, $t \in \mathbb{N}, t < T$, $|h(\mathbf{x}, t + 1, i) - h(\mathbf{x}, t, i)| \leq L(t)$.

At each algorithm iteration $k$, we make a decision $\mathbf{x}_k$, which we then apply to the system and get noisy measurements $y_k^i$ of the reward function and safety functions during the iteration. We use the index $k$ to refer to the algorithm iteration. Even though $k$ and $t$ might differ in principle, in practice we run one algorithm iteration $k$ for each time step $t$.

**Assumption 2.2.** Observations $y_k^i = h(\mathbf{x}_k, t, i) + \varepsilon_k^i$, $\forall i \in \mathcal{I}$, $t \in \mathbb{N}$ are perturbed by i.i.d. zero mean and $\sigma$-sub-Gaussian noise.

Based on the measurements, we compute the posterior GP and make the decision for the next time step. To start the exploration, an initial set of safe decisions is assumed to be available to the algorithm. To ensure that the safe set remains non-empty after the first iteration, it is necessary that the initial safety function values at every decision within the initial safe set are positive.

**Assumption 2.3.** An initial set $S_0 \subseteq \mathcal{X}$ of safe decisions is known and for all decisions $\mathbf{x} \in S_0$, we have $c_i(\mathbf{x}, 0) > 0$, $\forall i \in \mathcal{I}_c$.

Similar assumptions have also been made for the standard SAFEOPT algorithm [7] and are necessary to ensure feasibility of the exploration steps and to be able to identify new safe decision.

## 2.2 Safety Updates

To ensure safety, based on Assumption 2.1 and 2.2, we extend the definition of the confidence intervals from [7] so that, with high probability, they contain $f$ and $c_i$ using the posterior GP estimate given the data sampled so far. The confidence intervals for $h(\mathbf{x}, t, i)$ given training samples until iteration $k \geq 1$ are defined for all $\mathbf{x} \in \mathcal{X}$ and for all $i \in \mathcal{I}$ as

$$Q_k(\mathbf{x}, i) := \left[ \mu_{k-1}(\mathbf{x}, i) \pm \sqrt{\beta_k} \sigma_{k-1}(\mathbf{x}, i) \right], \tag{3}$$

where $\beta_k$ is a scalar that determines the desired confidence interval, $\mu_{k-1}(\mathbf{x}, i)$ and $\sigma_{k-1}(\mathbf{x}, i)$ are the posterior mean and standard deviation of $h(\mathbf{x}, t, i)$ inferred with $\mathcal{D}_k$, training samples till iteration $k$ [31]. The probability of the true function value $h$ lying within this interval depends on the choice of $\beta_k$ [8]. We provide more details for this choice in Section 2.4.

We now construct a tighter confidence interval for $h(\mathbf{x}, t, i)$ by using the sequence $\{Q_\tau(\mathbf{x}, i)\}_{\tau \leq k}$ instead of $Q_k(\mathbf{x}, i)$ alone. To this end, we recursively define for all $\mathbf{x} \in \mathcal{X}$ and for all $i \in \mathcal{I}$ the intersection

$$C_k(\mathbf{x}, i) := (C_{k-1}(\mathbf{x}, i) \oplus [-L(t-1), L(t-1)]) \cap Q_k(\mathbf{x}, i), \tag{4}$$

where $\oplus$ denotes the Minkowski sum, $C_0(\mathbf{x}, i)$ is $[L(0), \infty)$ for all $\mathbf{x} \in S_0$, $i \in \mathcal{I}_c$ and $\mathbb{R}$ otherwise. We use the lower bound $l_k(\mathbf{x}, i) := \min C_k(\mathbf{x}, i)$ and the upper bound $u_k(\mathbf{x}, i) := \max C_k(\mathbf{x}, i)$, to define the width of $C_k(\mathbf{x}, i)$

$$w_k(\mathbf{x}, i) := u_k(\mathbf{x}, i) - l_k(\mathbf{x}, i) \tag{5}$$

further used to update the safe set as well as pick the next decision to explore.

Based on the updated posterior and Lipschitz constants, we can update the safe set $S_k$ with the lower bounds $l_k$ and the previous safe set $S_{k-1}$ as

$$S_k = \cap_{i \in \mathcal{I}_c} \cup_{\mathbf{x} \in S_{k-1}} \left\{ \mathbf{x}' \in \mathcal{X} \mid l_k(\mathbf{x}, i) - L_{\mathbf{x}} d(\mathbf{x}, \mathbf{x}') - L(t) \geq 0 \right\}. \tag{6}$$

The set $S_k$ contains decisions that with high probability fulfill the safety constraints given the GP confidence intervals and the Lipschitz constants. In contrast to SAFEOPT, the safe set of TVSAFEOPT is allowed to shrink to adapt to the potential change of the safe region given the time-varying setting. However, the safe set might even become empty after the update. This is either because the safe region indeed becomes empty or because the updated safe set conservatively excludes all decisions with a lower bound of some safety function below $L$ to guarantee safety. In all these cases, if the updated safe set is empty, we terminate the algorithm.

## 2.3 Safe Exploration and Exploitation

With the safe set updated, the next challenge is to trade off between exploitation and expansion of the safe region. As in the standard SAFEOPT, the potential maximizers are those decisions, for which the upper confidence bound of the reward function is higher than the largest lower confidence bound

$$M_k = \left\{ \mathbf{x} \in S_k \mid u_k(\mathbf{x}, 0) \geq \max_{\mathbf{x}' \in S_k} l_k(\mathbf{x}', 0) \right\}. \tag{7}$$

To identify the potential expanders, $G_k$, containing all decisions that could potentially expand the safe set, we first quantify the potential enlargement of the current safe set after sampling a new decision $\mathbf{x}$. To do so, we define the function

$$e_k(\mathbf{x}) := |\{\mathbf{x}' \in \mathcal{X} \backslash S_k \mid \exists i \in \mathcal{I}_c : u_k(\mathbf{x}, i) - L_{\mathbf{x}} d(\mathbf{x}, \mathbf{x}') - L(t) \geq 0\}|, \tag{8}$$

where $|\cdot|$ refers to the cardinality of a set, and then update

$$G_k = \{\mathbf{x} \in S_k \mid e_k(\mathbf{x}) > 0\}. \tag{9}$$

At iteration $k$, TVSAFEOPT selects a decision $\mathbf{x}_k$ within the union of potential maximizers (7) and expanders (9)

$$\mathbf{x}_k = \operatorname*{arg\,max}_{\mathbf{x} \in G_k \cup M_k, i \in \mathcal{I}} w_k(\mathbf{x}, i), \tag{10}$$

**Algorithm 1** TVSAFEOPT

1: **Input:** Sample set $\mathcal{X}$
   GP priors for $f$, $c_i$
   Lipschitz constants $L_\mathbf{x}$ and $\{L(t)\}_{t \in \mathbb{N}, t < T}$
   Safe set seed $S_0$
2: $C_0(\mathbf{x}, i) \leftarrow [L(0), \infty)$, for all $\mathbf{x} \in S_0$, $i \in \mathcal{I}_c$
3: $C_0(\mathbf{x}, i) \leftarrow \mathbb{R}$, for all $\mathbf{x} \in \mathcal{X} \backslash S_0$, $i \in \mathcal{I}_c$
4: $C_0(\mathbf{x}, 0) \leftarrow \mathbb{R}$
5: Query a point $\mathbf{x}_0 \in S_0$, $y_0^i \leftarrow h(\mathbf{x}_0, 0, i) + \varepsilon_0^i$, $i \in \mathcal{I}$
6: $\mathcal{D}_0 = \{(\mathbf{x}_0, \mathbf{y}_0)\}$
7: **for** $k = 1, 2, \cdots, T$ **do**
8:   Calculate $Q_k(\mathbf{x}, i)$ as in (3), $\forall \mathbf{x} \in \mathcal{X}, \forall i \in \mathcal{I}$
9:   $C_k(\mathbf{x}, i) \leftarrow (C_{k-1}(\mathbf{x}, i) \oplus [-L(t-1), L(t-1)]) \cap Q_k(\mathbf{x}, i)$
10:   $S_k \leftarrow \cap_{i \in \mathcal{I}_c} \cup_{\mathbf{x} \in S_{t-1}} \{\mathbf{x}' \in \mathcal{X} \mid l_k(\mathbf{x}, i) - L_\mathbf{x} d(\mathbf{x}, \mathbf{x}') - L(t) \geq 0\}$
11:   **if** $S_k = \varnothing$ **then**
12:     break
13:   **end if**
14:   $M_k \leftarrow \{\mathbf{x} \in S_k \mid u_k(\mathbf{x}, 0) \geq \max_{\mathbf{x}' \in S_k} l_k(\mathbf{x}', 0)\}$
15:   $G_k \leftarrow \{\mathbf{x} \in S_k \mid e_k(\mathbf{x}) > 0\}$ with $e_k(\mathbf{x})$ from (8)
16:   $\mathbf{x}_k \leftarrow \arg\max_{\mathbf{x} \in G_k \cup M_k, i \in \mathcal{I}} w_k(\mathbf{x}, i)$
17:   $y_k^i \leftarrow h(\mathbf{x}_k, t, i) + \varepsilon_k^i$, $i \in \mathcal{I}$
18:   $\mathcal{D}_k = \mathcal{D}_{k-1} \cup \{(\mathbf{x}_k, \mathbf{y}_k)\}$
19: **end for**

with $w_k$ from (5). The objective of the greedy selection process in (10) is to take the most uncertain decision among the expanders $G_k$ and the maximizers $M_k$. The decision $\mathbf{x}_k$ is then applied to the system and after making observations of the reward and safety functions, $\mathbf{y}_k := (y_k^0, y_k^1, \ldots, y_k^m)$, we add $(\mathbf{x}_k, \mathbf{y}_k)$ to the training samples.

At any iteration, we can obtain an estimate for the current best decisions from

$$\hat{\mathbf{x}}_k = \arg\max_{\mathbf{x} \in S_k} l_k(\mathbf{x}, 0), \tag{11}$$

which returns the maximizer of the lower bound of the reward function within the current safe set.

## 2.4 Safety Guarantee

To provide safety guarantees, we need the confidence intervals in (3) to contain the safety functions with high probability for all iterations. Note that the parameter $\beta_k$ in (3) tunes the tightness of the confidence interval. The following lemma guides us to make a proper choice for $\beta_k$: This choice depends on the information capacity $\gamma_k^h$ associated with the kernel $\kappa$, namely is the maximal mutual information [33] we can obtain from the GP model of $h$ through $k$ noisy measurements $\hat{h}_{\mathbf{X}_k}$ at data points $\mathbf{X}_k := \{(\mathbf{x}_\tau \in \mathcal{X}, \tau, i_\tau \in \mathcal{I})\}_{\tau < k}$

$$\gamma_k^h := \max_{\mathbf{X}_k} I(\hat{h}_{\mathbf{X}_k}; h). \tag{12}$$

**Lemma 2.4.** *Assume that $h(\mathbf{x}, t, i)$ has RKHS norm associated with $\kappa$ bounded by $B > 0$ and that measurements are perturbed by $\sigma$-sub-Gaussian noise. Let the variable $\gamma_k^h$ be defined as in* (12). *For any $\delta \in (0, 1)$, let $\sqrt{\beta_k} = B + \sigma\sqrt{2\left(\gamma_{k \cdot |\mathcal{I}|}^h + 1 + \ln(1/\delta)\right)}$, then the following holds for all decisions $\mathbf{x} \in \mathcal{X}$, function indices $i \in \mathcal{I}$, and iterations $k \geq 1$ jointly with probability at least $1 - \delta$:*

$$|h(\mathbf{x}, t, i) - \mu_{k-1}(\mathbf{x}, i)| \leq \sqrt{\beta_k} \sigma_{k-1}(\mathbf{x}, i).$$

*Proof.* This lemma is a straightforward consequence of Lemma 1 of [16], a contextual extension of Lemma 4.1 of [8]. We can prove it by selecting time as the context and picking $\{t\}_{t \geq 1, t \in \mathbb{N}}$ as the context sequence. $\square$

Lemma 2.4 indicates that, by selecting $\beta_k$ properly, the confidence intervals $Q_k$ will w.h.p. contain the reward function and the safety functions. Due to this, they can be leveraged to provide theoretical guarantees for safety and optimality.

The following theorem provides a sufficient condition for safety of TVSAFEOPT.

**Theorem 2.5.** *Let Assumptions 2.1 - 2.3 hold, and let $\gamma_k^h$ be defined as in* (12)*. For any $\delta \in (0,1)$, let $\sqrt{\beta_k} = B + \sigma \sqrt{2 \left( \gamma_{k \cdot |\mathcal{I}|}^h + 1 + \ln(1/\delta) \right)}$, then TVSAFEOPT guarantees that with probability at least $1 - \delta$, for all $i \in \mathcal{I}_c$ and for all $t \geq 0$, and $\mathbf{x} \in S_k$ it holds $c_i(\mathbf{x}, t) \geq 0$.*

The proof builds on Lemma 2.4 to show first that for all $t \geq 0$, for all $i \in \mathcal{I}$ and for all $\mathbf{x} \in \mathcal{X}$, then $h(\mathbf{x}, t, i) \in C_k(\mathbf{x}, i)$ with high probability. Then using the recursive definition of the safe set from (6), we obtain w.h.p. $c_i(\mathbf{x}, t) \geq l_k(\mathbf{x}', i) - L_{\mathbf{x}} d(\mathbf{x}, \mathbf{x}') - L(t) \geq 0$, which concludes the proof. For details we refer the reader to Appendix B.

## 2.5 Near-Optimality Guarantee

In many safety critical real world applications, such as nuclear power plant operations, medical devices calibration, automated emergency response systems, the reward function is stationary most of the time. The problems are stationary until some changes happen and become stationary again when the systems reach new equilibria [34]. However, ensuring optimality is non-trivial even when the problem becomes stationary. Suppose the auxiliary function (2) becomes stationary in a time interval $[\underline{\phi}, \overline{\phi}]$, namely suppose there exist $\overline{\phi} > \underline{\phi} \geq 1$ such that $\forall t_1, t_2 \in [\underline{\phi}, \overline{\phi}], f(\mathbf{x}, t_1) = f(\mathbf{x}, t_2) =: \bar{f}(\mathbf{x})$ and $c(\mathbf{x}, t_1) = c(\mathbf{x}, t_2) =: \bar{c}(\mathbf{x})$, so that the optimization problem (1) becomes

$$\max_{\mathbf{x} \in \mathcal{X}} \bar{f}(\mathbf{x})$$
$$\text{subject to } \bar{c}_i(\mathbf{x}) \geq 0, \; i \in \mathcal{I}_c. \tag{13}$$

We first define the largest safe set expanded from a set $S$ subject to a measurement error $a$ within

- a single time step:
$$R_a(S) := S \cup \{\mathbf{x} \in \mathcal{X} \mid \forall i \in \mathcal{I}_c, \exists \mathbf{x}'_i \in S, s.t. \; \bar{c}_i(\mathbf{x}'_i) - L_{\mathbf{x}} d(\mathbf{x}, \mathbf{x}'_i) - a \geq 0\}$$

- $n$ time steps: $R_a^n(S) := \underbrace{R_a(R_a \ldots R_a(R_a(S)) \ldots)}_{n \text{ times}}$

- arbitrary time steps: $\bar{R}_a(S) := \lim_{n \to \infty} R_a^n(S)$

We also define $\bar{L}_t$ as an upper bound of the sum of all time Lipschitz constants, that is, $\sum_{\tau=0}^{T-1} L(\tau) \leq \bar{L}_t$.

We find it reasonable that a tight upper bound $\bar{L}_t$ can be provided when the underlying system slowly switches to the new stationarity condition.

Given these definitions, we are now in the position to provide optimality guarantees for TVSAFEOPT. In particular, we aim at comparing the found reward value $\bar{f}(\mathbf{x}_k)$ with the optimal reward value within the largest safe set obtained in ideal conditions with no measurement error, $\bar{R}_0(S_0)$. We also aim at providing TVSAFEOPT with an upper bound on the iterations needed to find a near-optimal solution. The following theorem states the optimality guarantee of TVSAFEOPT.

**Theorem 2.6.** *Let Assumptions 2.1 - 2.3 hold, let $\gamma_k^h$ be defined as in* (12) *and, for any $\delta \in (0,1)$, let $\sqrt{\beta_k} = B + \sigma \sqrt{2 \left( \gamma_{k \cdot |\mathcal{I}|}^h + 1 + \ln(1/\delta) \right)}$. Define $\hat{\mathbf{x}}_k$ as in (11), and, for any $\epsilon > 0$, let $k^*(\epsilon, \delta)$ be the smallest positive integer satisfying*

$$\frac{k^*}{\beta_{k^*} \gamma_{k^*}^h} \geq \frac{b_1 \left( \left| \bar{R}_0(S_0) \right| + 1 \right)}{\epsilon^2},$$

*where $b_1 = 8 / \log \left( 1 + \sigma^{-2} \right)$. Then, the TVSAFEOPT algorithm, applied to* (13)*, guarantees that, with probability at least $1 - \delta$, there exists $k \leq k^*$ such that*

$$\bar{f}(\hat{\mathbf{x}}_k) \geq \max_{\mathbf{x} \in \bar{R}_{\epsilon + \bar{L}_t}(S_0)} \bar{f}(\mathbf{x}) - \epsilon.$$

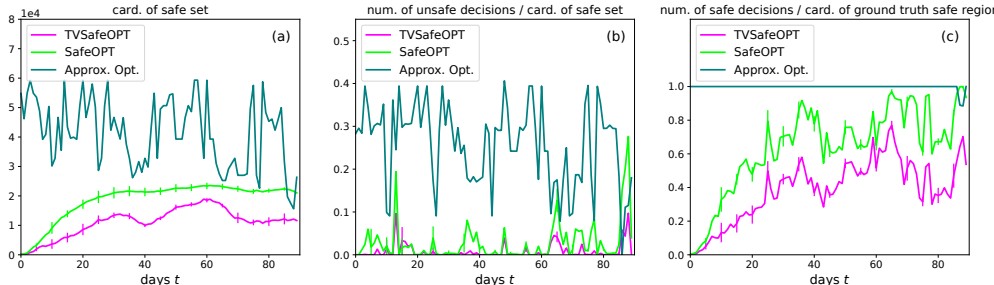

Figure 2: Comparison between TVSAFEOPT, SAFEOPT, and approximate optimization on the gas compressor case study, showing average of 10 repetitions with different initial sets. (a): The cardinality of the safe sets, (b): The ratio between the number of unsafe decisions in the safe sets and the cardinality of the safe sets, (c): The ratio between the number of safe decisions in the safe sets and the cardinality of the ground truth safe regions. TVSAFEOPT robustly shrinks its safe sets based on its observations and thus maintains much less violations in its safe sets than SAFEOPT and approximate optimization, at the cost of covering less of the ground truth safe region.

The proof consists in showing a decaying upper bound of uncertainty $w_k(\mathbf{x}, i) \leq \epsilon$ and exploiting local stationarity of (13) to provide bounds on the expansion of the safe set $S_k$. Details can be found in Appendix C.

## 3 Experiments

### 3.1 Synthetic Example

We first illustrate TVSAFEOPT on a synthetic two-dimensional time-varying optimization problem

$$\max_{x,y} - e^{x^2} - \log(1 + y^2) + 0.01t$$

$$\text{s.t.} \ \left[ x + 0.5 - 0.5 \left( 1 - \cos \frac{2\pi}{50} t \right) \cos \frac{\pi}{6} \right]^2 + \left[ y - 0.3 - 0.5 \left( 1 - \cos \frac{2\pi}{50} t \right) \sin \frac{\pi}{6} \right]^2 \leq 1.$$

Figure 1 compares the safe sets computed by TVSAFEOPT, ETSAFEOPT, and SAFEOPT at $t = 30$, $t = 100$ and $t = 170$. All algorithms start from the same singleton initial safe set $S_0 = \{(-0.5, 0.0)\}$. Implementation details are described in Appendix A. Figure 1 illustrates that the safe sets computed by TVSAFEOPT are contained in the ground truth safe regions while those computed by SAFEOPT and ETSAFEOPT have multiple violations. Due to the dependence on time of the example (Figure 3), the initial safe set becomes unsafe at $t = 30$, and $t = 170$. Taking the possible changes in time into consideration, TVSAFEOPT correctly identifies the possible unsafety of the initial safe set. In contrast, SAFEOPT always consider the initial safe set to be safe. Meanwhile, ETSAFEOPT correctly identifies the lack of safety of the initial safe set at $t = 30$, but fails at $t = 170$. This is because the event trigger is naturally insensitive to continuous changes. This toy example indicates that, in contrast to SAFEOPT and ETSAFEOPT, TVSAFEOPT safely adapts to the time changes of the optimization problem.

In this example, TVSAFEOPT and ETSAFEOPT overall find better reward function values than SAFEOPT, see Figure 3. The reward function value found by TVSAFEOPT is close to the optimal values when the reward function changes slowly, which supports Theorem 2.6.

Quantitative metrics are listed in Table 2. Taking SAFEOPT as a baseline, TVSAFEOPT and ETSAFEOPT achieves less violations, lower cumulative regret at the cost of covering less part of the safe region. Furthermore, TVSAFEOPT has little violations, and achieves larger coverage ratio than ETSAFEOPT. Meanwhile, its cumulative regret is just slightly higher than that of ETSAFEOPT.

Table 2: Synthetic example: comparison of TVSAFEOPT and ETSAFEOPT with respect to SAFEOPT, showing the average and the standard deviation results from five runs with different initial safe sets (chosen randomly from the feasible space).

|  | ETSAFEOPT | TVSAFEOPT |
| --- | --- | --- |
| Violations | -84.4% $\pm$ 1.7 % | -99.99% $\pm$ 0.01% |
| Coverage Ratio | -30.9% $\pm$ 2.9 % | -21.0% $\pm$ 1.3% |
| Cumulative Regret | -73.6% $\pm$ 14.7% | -66.9% $\pm$ 14.4% |

Table 3: Compressors case study: comparison of TVSAFEOPT and SAFEOPT with respect to Approximate Optimization, showing the average and the standard deviation results from 10 runs with different initial safe sets (chosen randomly from $[x_0 - 0.5/\sqrt{3}d, x_0 + 0.5/\sqrt{3}d]$ where $x_0$ is the initial safe seed and $d$ is the distance to the boundary of the feasible region). ETSAFEOPT is not included due to its high dependency on event detection methods, which are unavailable for compressor degradation.

|  | SAFEOPT | TVSAFEOPT |
| --- | --- | --- |
| Violations | -89.2% $\pm$ 4.2 % | -96.8% $\pm$ 1.0% |
| Coverage Ratio | -35.7% $\pm$ 2.6 % | -61.0% $\pm$ 1.3% |
| Cumulative Regret | +95.8% $\pm$ 32.3% | +178.3% $\pm$ 29.2% |

## 3.2 Gas Compressor Case Study

### 3.2.1 Problem Setup

We show the performance of the proposed algorithm in a compressor station with three identical compressors operating in parallel at the time-varying compressor head $H_t$ with time-varying power consumption at time $t$ (adapted from [35], details in Appendix A.3)

$$\min_{m_i} \sum_{i=1}^{N} \frac{1}{1 - d_{it}} \left( \alpha_1 + \alpha_2 \tilde{m}_i + \alpha_3 \tilde{H}_t + \alpha_4 \tilde{m}_i^2 + \alpha_5 \tilde{m}_i \tilde{H}_t + \alpha_6 \tilde{H}_t^2 \right) \tag{14}$$

$$\text{s.t.} \sum_{i=1}^{N} m_i \geq M_t \tag{15}$$

$$m_i \geq \beta_1 \bar{H}_t^2 + \beta_2 \bar{H}_t + \beta_3 \, , \forall i = 1, \ldots, N \tag{16}$$

$$m_i \geq \gamma_1 \bar{\bar{H}}_t^2 + \gamma_2 \bar{\bar{H}}_t + \gamma_3 \, , \forall i = 1, \ldots, N$$

$$m_i \leq \delta_1 \tilde{\tilde{H}}_t + \delta_2 \, , \forall i = 1, \ldots, N$$

$$m_i \leq \sigma_1 \tilde{\tilde{H}}_t^2 + \sigma_2 \tilde{\tilde{H}}_t + \sigma_3 \, , \forall i = 1, \ldots, N, \tag{17}$$

where the objective (14) corresponds to the power to run the station with $N$ compressors, here $N = 3$, affected by individual degradation $d_{it}$, $i = 1, l \ldots, N$. The station must also satisfy time-varying demand $M_t$ in (15). In practice, it is common to linearly approximate (16)-(17) with respect to the compressor head $H_t$ (dashed lines in Figure 4) [36].

### 3.2.2 Results

We compare the performance of TVSAFEOPT, SAFEOPT, and approximate optimization. ETSAFEOPT is not applicable to this case study because the magnitude of changes in the demand $M_t$, compressor head $H_t$, and degradation $d_{it}$ keeps triggering the event trigger and thus the maintained safe set becomes empty very quickly. Implementation details are described in Appendix A. Figure 2 compares the number of unsafe decisions in the safe sets calculated by TVSAFEOPT, SAFEOPT, and approximate optimization. We see that, by considering the uncertainty with respect to the decision variables, SAFEOPT maintains fewer unsafe decisions in its safe sets than the approximate optimization. However, SAFEOPT tends to expand its safe sets regardless of external changes. TVSAFEOPT further improves this based on SAFEOPT by taking into consideration the time-varying safety functions. TVSAFEOPT robustly shrinks its safe sets based on its observations and thus maintains much

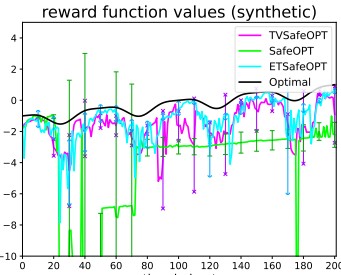
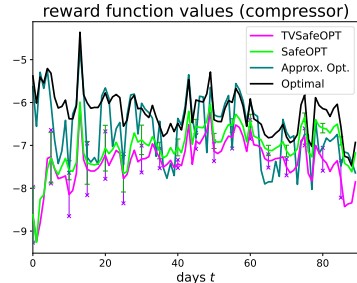

Figure 3: Comparison of reward functions from different methods with different initial safe sets, averaged over 5 runs for the synthetic example (left) and 10 runs for the compressor case study (right, indicating power in MW obtained from maximization of (14)), with error bars, with respect to the optimal values (black). In the synthetic example, TVSAFEOPT finds better reward function values than SAFEOPT, and similar to these of ETSAFEOPT. In the compressor case study, TVSAFEOPT finds lower reward function values than SAFEOPT, but guarantees fewer violations (Table 2 and 3) than either SAFEOPT or Approximate Optimization.

less violations in its safe sets than SAFEOPT (70.4%) and approximate optimization (96.8%). It achieves this at the cost of covering less of the ground truth safe region than SAFEOPT (39.3%) and Approximate Optimization (61.0%).

The right-hand side of Figure 3 shows that TVSAFEOPT preserves safety at the expense of optimality. In the compressor case study, TVSAFEOPT overall finds lower reward function values than SAFEOPT and approximate optimization, which is consistent with the fact that it covers a lower fraction of the ground truth safe regions and the reward function changes significantly between iterations. Because of its strong focus on safety, TVSAFEOPT deviates more from the ground truth. The cumulative regret of TVSAFEOPT is above the one of SAFEOPT by 42.1%, and the one of approximate optimization by 178.3%. This illustrates the trade-off between safety and optimality in the presence of strong uncertainties due to the varying reward and safety constraints. Quantitative metrics using Approximate Optimization as the baseline are listed in Table 3.

## 4   Limitations and Conclusion

**Limitations** The compressor case study demonstrated that TVSAFEOPT ensures safety at the expense of optimality if the stationarity assumption is not satisfied. The assumption about the local stationarity of the optimization problem (1) is thus the main limitation. Even though TVSAFEOPT demonstrates good empirical performance with respect to safety even when the problem is non-stationary, theoretical guarantee for its near-optimality in the non-stationary case warrant further investigation.

The need for obtaining the Lipschitz constants with respect to both $\mathbf{x}$ and $t$ in order to compute the safe set $S_k$ in (6) may prove limiting in real applications. To overcome this limitation, we propose practical modifications in Appendix A.1.

**Conclusions** We propose TVSAFEOPT algorithm, which extends SAFEOPT to handle time-varying optimization problems. In conclusion, TVSAFEOPT outperforms SAFEOPT in terms of adaptation to changes in time and maintains fewer unsafe decisions in its safe sets for time-varying problems. This is at the cost of covering less of the ground truth safe regions and may lead to poorer performance in terms of optimality.

We prove the safety guarantee for TVSAFEOPT in the general time-varying setting and prove its near-optimality guarantee for the case in which the optimization problem becomes stationary. The two theoretical results together guarantee that TVSAFEOPT is capable of safely transferring safety of the decisions into the future and, based on the transferred safe sets, it will find the near-optimal decision when the reward function stops changing. We show that TVSAFEOPT performs well in practice for the most general settings where both the reward function and the safety constraint are time-varying, both on synthetic data and for real case study on a gas compressor.

## Acknowledgments and Disclosure of Funding

Research supported by NCCR Automation, National Centre of Competence in Research, funded by the Swiss National Science Foundation (grant no. 180545). Marta Zagorowska also acknowledges funding from the Marie Curie Horizon Postdoctoral Fellowship project RELIC (grant no 101063948) for writing, revisions, and the compressor case study. Alisa Rupenyan acknowledges also support from the Johann Jakob Rieter foundation.

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

# A  Experiment Details

Experiments are conducted on an Intel i7-11370H CPU using Python 3.8.5. The implementation utilizes the following libraries: GPy 1.12.0, NumPy 1.22.0, and Matplotlib 3.5.0.

## A.1  Practical Modifications

In practice, Lipschitz constants are difficult to estimate. Thus, here we provide a Lipschitz-constant-free version of TVSAFEOPT algorithm by modifying (6) and (8).

The safe set is updated as all decisions with non-negative lower confidence bounds for the safety functions at the current iteration $k$, that is,

$$S_k = \{\mathbf{x} \in \mathcal{X} \mid \forall i \in \mathcal{I}_c, l_k(\mathbf{x}, i) \geq 0\}. \tag{18}$$

Furthermore, the expanders are intuitively defined as decisions within the current safe set such that, by evaluating any of the decisions, at least one decision outside the current safe set will be considered as safe, that is, $G_k = \{\mathbf{x} \in S_k \mid e_k(\mathbf{x}) > 0\}$, where $e_k(\mathbf{x})$ denotes the number of decisions outside $S_k$ that will be considered safe when evaluating $\mathbf{x}$. Instead of using Lipschitz constants, we define $e_k(\mathbf{x})$ using lower bound of auxiliary GP similar to the method by Berkenkamp et al. [37]

$$e_k(\mathbf{x}) = \left| \left\{ \mathbf{x}' \in \mathcal{D} \backslash S_k \mid \exists i \in \mathcal{I}_c : l_{k,(\mathbf{x}, u_k(\mathbf{x},i))} (\mathbf{x}', k+1, i) \geq 0 \right\} \right|,$$

where $l_{k,(\mathbf{x}, u_k(\mathbf{x},i))} (\mathbf{x}', k+1, i)$ denotes the lower bound of the function values at $\mathbf{x}$ and $t = k+1$ if $\mathbf{x}$ is evaluated at the $k$-th iteration and the upper bound is observed.

## A.2  Synthetic Example

The search space is $\mathcal{X} = [-2, 2]^2$, uniformly quantized into $100 \times 100$ points. Both algorithms start with the singleton initial safe set $\{(-0.5, 0.0)\}$. The measurements are perturbed by i.i.d. Gaussian noise $\mathcal{N}(0, 0.01^2)$.

The reward function is formulated as: $f(\mathbf{x}, t) = -e^{x^2} - \log(1 + y^2) + 0.01t$;

The safety function is formulated as: $c_1(\mathbf{x}, t) = 1 - \left[x + 0.5 - 0.5 \left(1 - \cos \frac{2\pi}{50}t\right) \cos \frac{\pi}{6}\right]^2 - \left[y - 0.3 - 0.5 \left(1 - \cos \frac{2\pi}{50}t\right) \sin \frac{\pi}{6}\right]^2$.

The hyperparameters of GPs in the synthetic case study are modelled as follows,

- TVSAFEOPT: The reward function and the safety function are modeled by independent GPs with zero mean and spatio-temporal kernel $\kappa((\mathbf{x}, t), (\mathbf{x}', t')) = \exp\left(-\frac{\|\mathbf{x} - \mathbf{x}'\|_2^2}{2\sigma_1^2}\right) \cdot \exp\left(-(\frac{(t - t')^2}{2\sigma_2^2})\right)$, where $\sigma_1 \equiv 1.0$, $\sigma_2 = 25.0$ for $f$, and $\sigma_2 = 15.0$ for $c_1$.

- SAFEOPT: The reward function and the safety function are modeled by independent GPs with zero mean and 2d Gaussian kernel $\kappa(\mathbf{x}, \mathbf{x}') = \exp\left(-\frac{\|\mathbf{x} - \mathbf{x}'\|_2^2}{2\sigma_3^2}\right)$, where $\sigma_3 \equiv 1.0$.

- ETSAFEOPT: Hyperparameters for GPs are the same as SAFEOPT. Besides, we choose the sentivity of the event trigger $\delta$ as 0.01.

## A.3  Compressor Case Study

Centrifugal compressors are often used in gas transport networks to deliver the required amount of gas by boosting the pressure in the pipelines. Organised as compressor stations with $N$ units, the compressors are often operated to minimise their power consumption $P$ while satisfying the demand $M_t$ and operating constraints, capturing how compressor head $H_t$ depends on the mass flow through the compressor [38, 39]:

- $\tilde{m}_i = \frac{m_i - 157.4}{34.37}$, $\tilde{H}_t = \frac{H_t - 1.016e5}{3.210e4}$, $\alpha_1 = 1.979e7$, $\alpha_2 = 5.274e6$, $\alpha_3 = 5.375e6$, $\alpha_4 = 6.055e5$, $\alpha_5 = 5.718e6$, $\alpha_6 = 3.319e5$

- $\bar{H}_t = \frac{H_t - 1.235e5}{3.764e4}$, $\beta_1 = -1.953$, $\beta_2 = 16.86$, $\beta_3 = 118.1$

- $\bar{\bar{H}}_t = \frac{H_t - 6.152e4}{7002}$, $\gamma_1 = -1.516$, $\gamma_2 = -11.12$, $\gamma_3 = 116.9$

- $\tilde{\bar{H}}_t = \frac{H_t - 8.706e4}{5.289e4}$, $\delta_1 = 73.21$, $\delta_2 = 183.7$

- $\tilde{\tilde{H}}_t = \frac{H_t - 1.572e5}{2.044e4}$, $\sigma_1 = -7.260$, $\sigma_2 = -29.65$, $\sigma_3 = 204.4$

The compressor case study has been adapted from [35]. The data for the demand, compressor head, and degradation for the three compressors were obtained from [40] (Creative Commons Attribution NonCommercial Licence).

Individual characteristics of compressors in (16)-(17) are called *compressor maps* (Figure 4). The operating area for a compressor is defined by minimal and maximal speed of the compressor and its mechanical properties. The operating area can be obtained from compressors maps delivered by the manufacturer of the compressor, or estimated during the operation [41]. However, estimation would require collecting datapoints close to the boundary of the operating area, which may be unavailable due to safety consideration [42, 43]. Using safe learning has the potential to improve the operation of the station because it enables safe exploration of the unknown operating area of individual compressors.

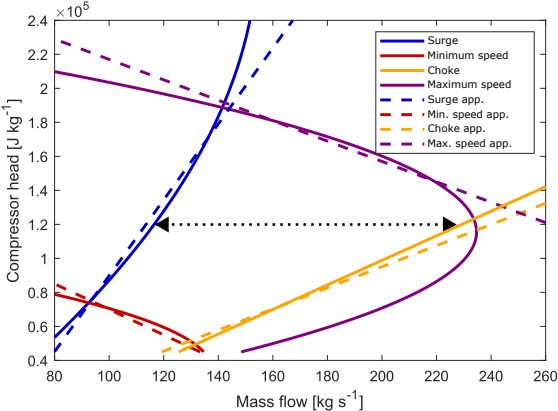

Figure 4: Ground truth (solid) and linear approximation (dashed) of the operating area from compressor maps, adapted from [44, 35]. For a given compressor head at time $t$ (dotted horizontal line for $H_t = 120000 \, \text{J kg}^{-1}$), the mass flow $m_{it}$ through the $i$-th compressor is required to be between minimum speed (red) and surge (blue) lines, and maximum speed (violet) and choke (yellow) lines

Furthermore, varying operating conditions and demand often lead to compressor degradation $d_{it}$ (Figure 5), over time increasing power consumption (14) of the entire compressor station [45]. Capturing the time-varying aspect of compressor degradation is a subject of research (e.g. [46–48]) but limited availability of measured degradation data presents a challenge [49].

For convenience, the optimization variables are scaled by a factor $K = 200$, that is, $\mathbf{x} = (m_1, m_2, m_3)/K$. The search space is $\mathcal{X} = [50.0/K, 250.0/K]^3$, uniformly quantized into $60 \times 60 \times 60$ points. Both algorithms start with the singleton initial safe set $\{(M_0, M_0.M_0)/3K\}$. The measurements are perturbed by i.i.d. Gaussian noise $\mathcal{N}(0, 0.01^2)$.

The reward function is formulated as:

$$f(\mathbf{x}, t) = -\sum_{i=1}^{3} \frac{1}{(1 - d_{it}) \cdot 10^7} \left( \alpha_1 + \alpha_2 \tilde{m}_i + \alpha_3 \tilde{H}_t + \alpha_4 \tilde{m}_i^2 + \alpha_5 \tilde{m}_i \tilde{H}_t + \alpha_6 \tilde{H}_t^2 \right)$$

The safety functions are formulated as $c_i(x, t) \geq 0$, $i = 1, \ldots, 7$, with:

- $c_1(\mathbf{x}, t) = x_1 - L_t$

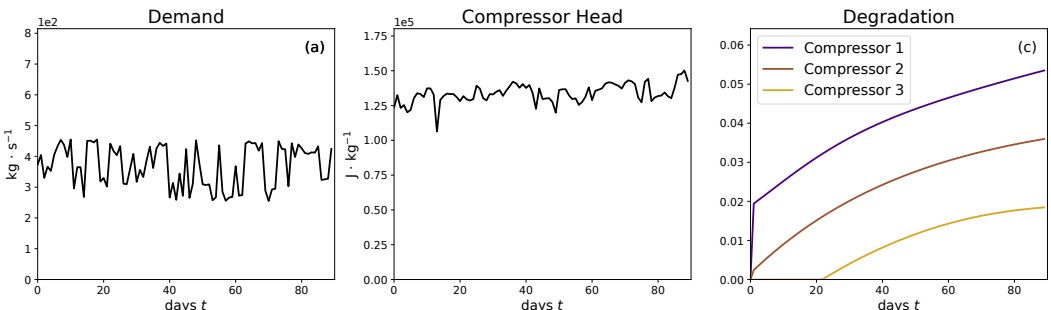

Figure 5: Visualization of demand (a), compressor head (b), and degradation for the compressors (c) changing with time.

- $c_2(\mathbf{x}, t) = U_t - x_1$
- $c_3(\mathbf{x}, t) = x_2 - L_t$
- $c_4(\mathbf{x}, t) = U_t - x_2$
- $c_5(\mathbf{x}, t) = x_3 - L_t$
- $c_6(\mathbf{x}, t) = U_t - x_3$
- $c_7(\mathbf{x}, t) = x_1 + x_2 + x_3 - 0.67 M_t / K,$

where $L_t = \max\{\beta_1 \bar{H}_t^2 + \beta_2 \bar{H}_t + \beta_3, \gamma_1 \bar{\bar{H}}_t^2 + \gamma_2 \bar{\bar{H}}_t + \gamma_3\}/K$, $U_t = \min\{\delta_1 \tilde{H}_t + \delta_2, \sigma_1 \tilde{\tilde{H}}_t^2 + \sigma_2 \tilde{\tilde{H}}_t + \sigma_3\}/K$.

The hyperparameters of GPs in the compressor case study are modelled as follows,

- TVSAFEOPT: The reward function and the safety functions are modeled by independent GPs with zero mean and spatio-temporal kernel $\kappa((\mathbf{x}, t), (\mathbf{x}', t')) = \exp\left(-\frac{\|\mathbf{x}-\mathbf{x}'\|_2^2}{2\sigma_1^2}\right) \cdot \exp\left(-\left(\frac{t-t'}{2\sigma_2^2}\right)^2\right)$, where $\sigma_1 \equiv 1.0$, $\sigma_2 = 80.0$ for $f$ and $c_1$ - $c_6$, and $\sigma_2 = 70.0$ for $c_7$.
- SAFEOPT: The reward function and the safety functions are modeled by independent GPs with zero mean and 3d Gaussian kernel $\kappa(\mathbf{x}, \mathbf{x}') = \exp\left(-\frac{\|\mathbf{x}-\mathbf{x}'\|_2^2}{2\sigma_3^2}\right)$, where $\sigma_3 \equiv 1.0$.

As for approximate optimization, the r.h.s. of (16) - (17) are linearly approximated as follows:

- Surge line: $\beta_1 \bar{H}_t^2 + \beta_2 \bar{H}_t + \beta_3 \approx 4.481e - 4 \cdot H_t + 59.76$
- Min. speed line: $\gamma_1 \bar{\bar{H}}_t^2 + \gamma_2 \bar{\bar{H}}_t + \gamma_3 \approx -1.333e - 3 \cdot H_t + 193.3$
- Choke line: $\delta_1 \tilde{H}_t + \delta_2 \approx 1.611e - 3 \cdot H_t + 46.77$
- Max. speed line: $\sigma_1 \tilde{\tilde{H}}_t^2 + \sigma_2 \tilde{\tilde{H}}_t + \sigma_3 \approx -1.667e - 3 \cdot H_t + 461.7$

# B   Proof of Safety Guarantee

Note all following lemmas hold for any $\delta \in (0, 1)$, and $S_0$, such that $\varnothing \subsetneq S_0 \subseteq \mathcal{X}$.

First, we want to show that the intersected confidence interval $C_k$ in (4) w.h.p. contains the reward function and safety functions $h(\mathbf{x}, t, i)$ as in (2).

**Lemma B.1.** *Let* $\sqrt{\beta_k} = B + \sigma\sqrt{2\left(\gamma^h_{k \cdot |\mathcal{I}|} + 1 + \ln(1/\delta)\right)}$, *with* $\gamma^h_k$ *defined as in (12) and* $C_k(\mathbf{x}, i)$ *defined as in (4), then the following holds with probability at least* $1 - \delta$ :

$$h(\mathbf{x}, t, i) \in C_k(\mathbf{x}, i) \qquad \forall t \geq 0, \forall i \in \mathcal{I}, \forall \mathbf{x} \in \mathcal{X},$$

*Proof by induction.*
If $t = 0$, by Assumption 2.3 and the definition of $C_k$ in (4), then $h(\mathbf{x}, 0, i) \in C_0(\mathbf{x}, i)$, for all $i \in \mathcal{I}$ and for all $\mathbf{x} \in \mathcal{X}$.

Suppose, for any $t = \tau \geq 0$, that $h(\mathbf{x}, \tau, i) \in C_\tau(\mathbf{x}, i)$, then for $t = \tau + 1$, from the Lipschitz continuity of $h$, $|h(\mathbf{x}, \tau + 1, i) - h(\mathbf{x}, \tau, i)| \leq L(\tau)$, it holds that $h(\mathbf{x}, \tau + 1, i) \in C_\tau(\mathbf{x}, i) \oplus [-L(\tau), L(\tau)]$.

Moreover, by Lemma 2.4 and (3) we have that $h(\mathbf{x}, \tau + 1, i) \in Q_{\tau+1}(\mathbf{x}, i)$.

Thus, $h(\mathbf{x}, \tau + 1, i) \in (C_\tau(\mathbf{x}, i) \oplus [-L(\tau), L(\tau)]) \cap Q_{\tau+1}(\mathbf{x}, i) = C_{\tau+1}(\mathbf{x}, i)$, $\forall i \in \mathcal{I}$, $\forall \mathbf{x} \in \mathcal{X}$.

Therefore, for all $t \geq 0$, for all $i \in \mathcal{I}$ and for all $\mathbf{x} \in \mathcal{X}$ we have that $h(\mathbf{x}, t, i) \in C_k(\mathbf{x}, i)$, and this concludes the proof. $\qquad\square$

We are now ready to prove Theorem 2.5 that provides a sufficient condition for TVSAFEOPT to ensure safety embedded in the constraints $c_i(\mathbf{x}, t) \geq 0$, for all $i \in \mathcal{I}_c$.

*Proof of Theorem 2.5.*
If $t = 0$, by definition of $S_0$, one has $c_i(\mathbf{x}, t) = c_i(\mathbf{x}, 0) \geq L(0) \geq 0, \forall i \in \mathcal{I}_c, \forall \mathbf{x} \in S_0$.

For any $t \geq 1$, $\forall \mathbf{x} \in S_t$, by recursive definition of $S_k$ in (6), $\forall i \in \mathcal{I}_c$, there exists $\mathbf{x}' \in S_{t-1}$, s.t. $l_k(\mathbf{x}', i) - L_{\mathbf{x}}d(\mathbf{x}, \mathbf{x}') - L(t) \geq 0$. Then, $\forall i \in \mathcal{I}_c$

$$
\begin{aligned}
&c_i(\mathbf{x}, t) \\
\geq\, &c_i(\mathbf{x}', t) - L_{\mathbf{x}}d(\mathbf{x}, \mathbf{x}') && \text{by Lipschitz continuity with } \mathbf{x} \\
\geq\, &l_k(\mathbf{x}', i) - L_{\mathbf{x}}d(\mathbf{x}, \mathbf{x}') && \text{by Lemma B.1} \\
\geq\, &l_k(\mathbf{x}', i) - L_{\mathbf{x}}d(\mathbf{x}, \mathbf{x}') - L(t) \\
\geq\, &0
\end{aligned}
$$

and this concludes the proof. $\qquad\square$

# C Proof of Near-Optimality Guarantee

The proof of near optimality consists in two parts: i) bounding the uncertainty and ii) bounding the expansion of the safe set.

## C.1 Bounding the Uncertainty

We first derive a decaying upper bound of uncertainty for TVSAFEOPT. In this way we can ensure the uncertainty of the reward function and safety functions to drop below a desired threshold.

**Lemma C.1.** *Define $b_1 := 8/\log\left(1 + \sigma^{-2}\right) \in \mathbb{R}$, and $\gamma_k^h$ as in (12). For any $k > k_0 \geq 1$, there exists $k' \in (k_0, k]$, such that the following holds for all $i \in \mathcal{I}$:*

$$w_{k'}(\mathbf{x}_{k'}, i) \leq \sqrt{\frac{b_1 \beta_k \gamma_k^h}{k - k_0}},$$

*Proof.*
Let $i_\tau := \arg\max_{i \in \mathcal{I}} w_\tau(\mathbf{x}_\tau, i)$, where $\mathbf{x}_\tau = \arg\max_{\mathbf{x} \in G_\tau \cup M_\tau} \max_{i \in \mathcal{I}} w_\tau(\mathbf{x}, i)$. For all $i \in \mathcal{I}$, $k_0 < k$, there exists $k' \in (k_0, k]$:

$$w_{k'}(\mathbf{x}_{k'}, i)$$

$$\leq \frac{1}{k - k_0} \sum_{\tau = k_0 + 1}^{k} w_\tau(\mathbf{x}_\tau, i_\tau)$$

$$\overset{(a)}{\leq} \frac{2}{k - k_0} \sum_{\tau = k_0 + 1}^{k} \sqrt{\beta_\tau} \sigma_{\tau-1}(\mathbf{x}_\tau, i_\tau)$$

$$\leq \frac{2\sqrt{\beta_k}}{k - k_0} \sum_{\tau = k_0 + 1}^{k} \sigma_{\tau-1}(\mathbf{x}_\tau, i_\tau)$$

$$\overset{(b)}{\leq} \sqrt{\frac{4\beta_k}{k - k_0} \sum_{\tau = k_0 + 1}^{k} \sigma_{\tau-1}^2(\mathbf{x}_\tau, i_\tau)}$$

$$\overset{(c)}{\leq} \sqrt{\frac{b_1 \beta_k}{k - k_0} \frac{1}{2} \sum_{\tau = k_0 + 1}^{k} \log(1 + \sigma^{-2}\sigma_{\tau-1}^2(\mathbf{x}_\tau, i_\tau))}$$

$$\overset{(d)}{\leq} \sqrt{\frac{b_1 \beta_k}{k - k_0} \frac{1}{2} \sum_{\tau = k_0 + 1}^{k} \log(1 + \sigma^{-2}\sigma_{\tau-1}'^2(\mathbf{x}_\tau, i_\tau))}$$

$$\overset{(e)}{=} \sqrt{\frac{b_1 \beta_k I(\hat{h}_{\mathbf{X}_k}; h)}{k - k_0}}$$

$$\overset{(f)}{\leq} \sqrt{\frac{b_1 \beta_k \gamma_k^h}{k - k_0}}$$

(a): Definition of $w_k$ in (5),

(b): From the fact that the quadratic mean upper bounds the arithmetic mean,

(c): $\sigma_{\tau-1}^2(\mathbf{x}_\tau, i_\tau) \leq k\left((\mathbf{x}_\tau, \tau, i_\tau), (\mathbf{x}_\tau, \tau, i_\tau)\right) \leq 1$ by Assumption 2.1, and the fact that $a \leq \frac{b_1}{8} \log(1 + \sigma^{-2}a)$, $\forall a \in [0, 1]$,

(d): $\sigma_{\tau-1}'(\mathbf{x}, i)$ denotes the posterior standard deviation of $h(\mathbf{x}, \tau, i)$ inferred by observations at $\mathbf{X}_\tau := \{(\mathbf{x}_j, j, i_j)\}_{j < \tau}$. Since $\{(\mathbf{x}_j, j, i_j)\}_{j =< \tau} \subsetneq \{(\mathbf{x}_j, j)\}_{j < \tau} \times \mathcal{I}$, then $\sigma_{\tau-1}(\mathbf{x}_\tau, i_\tau) \leq \sigma_{\tau-1}'(\mathbf{x}_\tau, i_\tau)$,

(e): From [19, Lemma 5.3],

(f): Definition of $\gamma_k^h$ (12).

$\square$

**Corollary C.2.** *Given $b_1 := 8/\log\left(1 + \sigma^{-2}\right) \in \mathbb{R}$, take $T_k$ as the smallest positive integer satisfying $\frac{T_k}{\beta_{k+T_k}\gamma_{k+T_k}^h} \geq \frac{b_1}{\epsilon^2}$. Then, there exists $k' \in (k, k + T_k]$, such that for any $\mathbf{x} \in G_{k'} \cup M_{k'}$, and for all $i \in \mathcal{I}$ it holds that*

$$w_{k'}(\mathbf{x}, i) \leq \epsilon.$$

## C.2 Bounding the Expansion of the Safe Set

All following lemmas hold for any $\delta \in (0, 1), \epsilon > 0$ and $S_0$, such that $\varnothing \subsetneq S_0 \subseteq \mathcal{X}$.

To facilitate the theoretical analysis, we define $\forall \mathbf{x} \in \mathcal{X}, \forall i \in \mathcal{I}$:

$$\begin{cases} \tilde{l}_k(\mathbf{x}, i) := \max\{\tilde{l}_{k-1}(\mathbf{x}, i), \mu_{k-1}(\mathbf{x}, i) - \beta_k^{1/2}\sigma_{k-1}(\mathbf{x}, i)\}, & k \geq 1 \\ \tilde{l}_0(\mathbf{x}, i) := l_0(\mathbf{x}, i) \end{cases} \tag{19}$$

Remember that, from (4), we can derive $\forall \mathbf{x} \in \mathcal{X}, \forall i \in \mathcal{I}$:

$$l_k(\mathbf{x}, i) = \max\{l_{k-1}(\mathbf{x}, i) - L(t-1), \ \mu_{k-1}(\mathbf{x}, i) - \beta_k^{1/2}\sigma_{k-1}(\mathbf{x}, i)\} \tag{20}$$

Therefore, $\tilde{l}_k$ can be viewed as updating $l_k$ with $L(t) \equiv 0$. With a slight abuse of notation, we omit arguments $\mathbf{x}$ and $i$ when not ambiguous.

**Lemma C.3.** *The following holds for any $k \geq 1, \forall \mathbf{x} \in \mathcal{X}, \forall i \in \mathcal{I}$:*

*(i) $l_k(\mathbf{x}, i) \geq l_{k-1}(\mathbf{x}, i) - L(t-1)$*

*(ii) $\tilde{l}_k(\mathbf{x}, i) \geq \tilde{l}_{k-1}(\mathbf{x}, i)$*

*(iii) $l_k(\mathbf{x}, i) \leq \tilde{l}_k(\mathbf{x}, i)$*

*(iv) $\tilde{l}_k(\mathbf{x}, i) - \bar{L}_t \leq l_k(\mathbf{x}, i)$*

*Proof.*

(i) Direct consequence of (20).

(ii) Direct consequence of (19).

(iii) We proceed by induction. Suppose $l_\tau \leq \tilde{l}_\tau$, then $l_\tau - L(\tau) \leq \tilde{l}_\tau$, thus according to (20), $l_{\tau+1} = \max\{l_\tau - L(\tau), \ \mu_\tau - \beta_{\tau+1}^{1/2}\sigma_\tau\} \leq \max\{\tilde{l}_\tau, \ \mu_\tau - \beta_{\tau+1}^{1/2}\sigma_\tau\} = \tilde{l}_{\tau+1}$, from which it follows $l_k(\mathbf{x}, i) \leq \tilde{l}_k(\mathbf{x}, i)$.

(iv) We proceed by induction. Suppose $l_\tau \geq \tilde{l}_\tau - \sum_{k=0}^{\tau-1} L(k)$.

If $\mu_\tau - \beta_{\tau+1}^{1/2}\sigma_\tau > \tilde{l}_\tau$, then $l_{\tau+1} \overset{(20)}{=} \mu_\tau - \beta_{\tau+1}^{1/2}\sigma_\tau \overset{(19)}{=} \tilde{l}_{\tau+1} \geq \tilde{l}_{\tau+1} - \sum_{k=0}^{\tau} L(k)$.

If $\mu_\tau - \beta_{\tau+1}^{1/2}\sigma_\tau < l_\tau - L(\tau)$, then $l_{\tau+1} \overset{(20)}{=} l_\tau - L(\tau) \geq \tilde{l}_\tau - \sum_{k=0}^{\tau-1} L(k) - L(\tau) = \tilde{l}_{\tau+1} - \sum_{k=0}^{\tau} L(k)$.

Otherwise, $l_{\tau+1} \overset{(20)}{=} \mu_\tau - \beta_{\tau+1}^{1/2}\sigma_\tau \geq l_\tau - L(\tau) \geq \tilde{l}_\tau - \sum_{k=0}^{\tau-1} L(k) - L(\tau) = \tilde{l}_{\tau+1} - \sum_{k=0}^{\tau} L(k)$.

To summarize, $l_k \geq \tilde{l}_k - \sum_{k=0}^{t-1} L(k) \geq \tilde{l}_k - \bar{L}_\mathrm{t}$

$\square$

Lemma C.3 allows us to define auxiliary safe sets based on $\tilde{l}_k$ such that they are contained in $S_k$. Furthermore, due to the monotonicity of $\tilde{l}_k$, we can prove the auxiliary safe sets never shrink, which will play a fundamental role in studying their convergence property and provide near-optimality guarantee of TVSAFEOPT.

Based on (19), we further define:

$\overline{S}_k := \{\mathbf{x} \in \mathcal{X} \mid \forall i \in \mathcal{I}_c, \exists \mathbf{x}'_i \in \overline{S}_{t-1}, s.t. \; \tilde{l}_k(\mathbf{x}'_i, i) - L_\mathbf{x} d(\mathbf{x}, \mathbf{x}'_i) \geq 0\}$

$\underline{S}_k := \{\mathbf{x} \in \mathcal{X} \mid \forall i \in \mathcal{I}_c, \exists \mathbf{x}'_i \in \underline{S}_{t-1}, s.t. \; \tilde{l}_k(\mathbf{x}'_i, i) - L_\mathbf{x} d(\mathbf{x}, \mathbf{x}'_i) - \bar{L}_\mathrm{t} \geq 0\}$

$\overline{S}_0 = \underline{S}_0 = S_0$

Remember $S_k = \{\mathbf{x} \in \mathcal{X} \mid \forall i \in \mathcal{I}_c, \exists \mathbf{x}'_i \in S_{t-1}, s.t. \; l_k(\mathbf{x}'_i, i) - L_\mathbf{x} d(\mathbf{x}, \mathbf{x}'_i) - L(t) \geq 0\}$. Thus, $\overline{S}_k = \underline{S}_k = S_k$ if and only if $L(t) \equiv 0$.

The following lemma proves that $\underline{S}_k$ never shrinks, and that $\underline{S}_k$ and $\overline{S}_k$ are a subset and a superset for $S_k$, respectively.

**Lemma C.4.** *The following holds for any $t \geq 1$:*

(i) $\underline{S}_{t-1} \subseteq \underline{S}_k$

(ii) $\underline{S}_k \subseteq S_k \subseteq \overline{S}_k$

*Proof.*

(i) We refer the reader to [8, Lemma 7.1].

(ii) We proceed by induction. Suppose $\underline{S}_\tau \subseteq S_\tau \subseteq \overline{S}_\tau$.

For all $\mathbf{x} \in S_{\tau+1}$, and for all $i \in \mathcal{I}_c$, there exists $\mathbf{x}'_i \in S_\tau \subseteq \overline{S}_\tau$, s.t. $\tilde{l}_\tau(\mathbf{x}'_i, i) - L_\mathbf{x} d(\mathbf{x}, \mathbf{x}'_i) \geq l_\tau(\mathbf{x}'_i, i) - L_\mathbf{x} d(\mathbf{x}, \mathbf{x}'_i) - L(\tau) \geq 0$, hence $\mathbf{x} \in \overline{S}_{\tau+1}$ as well. Therefore, $S_\tau \subseteq \overline{S}_\tau \; \forall \tau$.

For all $\mathbf{x} \in \underline{S}_{\tau+1}$, and for all $i \in \mathcal{I}_c$, there exists $\mathbf{x}'_i \in \underline{S}_\tau \subseteq S_\tau$, s.t. $l_\tau(\mathbf{x}'_i, i) - L_\mathbf{x} d(\mathbf{x}, \mathbf{x}'_i) - L(\tau) \geq \tilde{l}_\tau(\mathbf{x}'_i, i) - \sum_{k=0}^{\tau-1} L(k) - L_\mathbf{x} d(\mathbf{x}, \mathbf{x}'_i) - L(\tau) = \tilde{l}_\tau(\mathbf{x}'_i, i) - L_\mathbf{x} d(\mathbf{x}, \mathbf{x}'_i) - \sum_{k=0}^{\tau} L(k) \geq \tilde{l}_\tau(\mathbf{x}'_i, i) - L_\mathbf{x} d(\mathbf{x}, \mathbf{x}'_i) - \bar{L}_\mathrm{t} \geq 0$, thus $\mathbf{x} \in S_{\tau+1}$. Therefore, $\underline{S}_\tau \subseteq S_\tau, \forall \tau$. From which we conclude $\underline{S}_\tau \subseteq S_\tau \subseteq \overline{S}_\tau, \forall \tau$.

$\square$

**Note:** Where needed in the following lemmas, we assume $b_1$ and $T_k$ are defined as in Lemma C.1 and Corollary C.2

**Lemma C.5** (Lemma 7.4 in [8])**.** *For any $k \geq 1$, $a > 0$, if $\bar{R}_a(S_0) \setminus \underline{S}_k \neq \varnothing$, then $R_a(\underline{S}_k) \setminus \underline{S}_k \neq \varnothing$.*

The following lemma provides a sufficient condition for the expansion of the auxiliary safe set $\underline{S}_k$.

**Lemma C.6.** *For any $t \geq 1$, if $\bar{R}_{\bar{L}_\mathrm{t}+\epsilon}(S_0) \setminus \underline{S}_k \neq \varnothing$, then, with probability at least $1 - \delta$, it holds that $\underline{S}_{k+T_k} \supsetneq \underline{S}_k$.*

*Proof.*
Similar to the proof of [8, Lemma 7.5].

By Lemma C.5, we get that, $R_{\bar{L}_\mathrm{t}+\epsilon}(\underline{S}_k) \setminus \underline{S}_k \neq \varnothing$. Equivalently, $\exists \mathbf{x} \in R_{\bar{L}_\mathrm{t}+\epsilon}(\underline{S}_k) \setminus \underline{S}_k$ which implies that, for all $i \in \mathcal{I}_c$,

$$\exists \mathbf{z}_i \in \underline{S}_k : \bar{c}_i(\mathbf{z}_i) - L_\mathbf{x} d(\mathbf{z}_i, \mathbf{x}) - \bar{L}_\mathrm{t} - \epsilon \geq 0$$

Now assume, to the contrary, that $\underline{S}_{k+T_k} = \underline{S}_k$. Thus, $\forall k' \in (k, k+T_k]$, $\mathbf{x} \in \mathcal{D}\backslash\underline{S}_{k'}$, and $\forall i \in \mathcal{I}_c$, $\mathbf{z}_i \in \underline{S}_{k'}$.

$$u_{k'}(\mathbf{z}_i, i) - L_{\mathbf{x}}d(\mathbf{z}_i, \mathbf{x}) - L(k')$$
$$\geq \bar{c}_i(\mathbf{z}_i) - L_{\mathbf{x}}d(\mathbf{z}_i, \mathbf{x}) - L(k') \qquad \text{by Lemma B.1}$$
$$\geq \bar{c}_i(\mathbf{z}_i) - L_{\mathbf{x}}d(\mathbf{z}_i, \mathbf{x}) - \bar{L}_{\mathrm{t}} - \epsilon$$
$$\geq 0$$

Therefore, by definition (8), $e_{k'}(\mathbf{z}_i) > 0$, which implies $\mathbf{z}_i \in G_{k'}$, $\forall k' \in (k, k+T_k]$, $\forall i \in \mathcal{I}_c$.

Therefore, we know that there exists $k' \in (k, k+T_k]$, for all $i \in \mathcal{I}_c$, $w_{k'}(\mathbf{z}_i, i) \leq \epsilon$. (Corollary C.2) Hence, for all $i \in \mathcal{I}_c$,

$$\tilde{l}_{k'}(\mathbf{z}_i, i) - L_{\mathbf{x}}d(\mathbf{z}_i, \mathbf{x})$$
$$\geq \bar{c}_i(\mathbf{z}_i) - w_{k'}(\mathbf{z}_i, i) - L_{\mathbf{x}}d(\mathbf{z}_i, \mathbf{x}) \qquad \text{by Lemma B.1}$$
$$\geq \bar{c}_i(\mathbf{z}_i) - \epsilon - L_{\mathbf{x}}d(\mathbf{z}_i, \mathbf{x})$$
$$\geq \bar{L}_{\mathrm{t}}$$

This means $\mathbf{x} \in \underline{S}_{k'} = \underline{S}_k$, which leads to a contradiction.

$\square$

The following lemma gives a superset for the auxiliary safe set $\underline{S}_k$.

**Lemma C.7.** $\underline{S}_k \subseteq \bar{R}_{\bar{L}_{\mathrm{t}}}(S_0)$ *with probability at least* $1 - \delta$.

*Proof by induction.*
$\underline{S}_0 = S_0 \subseteq \bar{R}_{\bar{L}_{\mathrm{t}}}(S_0)$

Suppose $\underline{S}_\tau \subseteq \bar{R}_{\bar{L}_{\mathrm{t}}}(S_0)$.

For all $\mathbf{x} \in \underline{S}_{\tau+1}$ and for all $i \in \mathcal{I}_c$ there exists $\mathbf{x}_i' \in \underline{S}_\tau$, s.t. $\bar{c}_i(\mathbf{x}_i') - L_{\mathbf{x}}d(\mathbf{x}, \mathbf{x}_i') - \bar{L}_{\mathrm{t}} \overset{(a)}{\geq} \tilde{l}_k(\mathbf{x}_i', i) - L_{\mathbf{x}}d(\mathbf{x}, \mathbf{x}_i') - \bar{L}_{\mathrm{t}} \geq 0$.

(a): Lemma B.1.

Thus, $\underline{S}_{\tau+1} \subseteq R_{\bar{L}_{\mathrm{t}}}(\underline{S}_\tau) \subseteq \bar{R}_{\bar{L}_{\mathrm{t}}}(S_0)$ $\square$

**Lemma C.8** (Lemma 7.8 in [8])**.** *Let $k^*$ be the smallest integer, such that $k^* \geq \left|\bar{R}_{\bar{L}_{\mathrm{t}}}(S_0)\right| T_{k^*}$. Then, there exists $k_0 \leq k^*$, such that $\underline{S}_{k_0+T_{k_0}} = \underline{S}_{k_0}$.*

Lemma C.8 together with Lemma C.6, and Lemma C.7 entail convergence of $\underline{S}_k$ within $k^*$ time steps, which ultimately leads us to the near-optimality of TVSAFEOPT when the problem becomes stationary.

**Lemma C.9.** *For any $k \geq 1$, if $\underline{S}_{k+T_k} = \underline{S}_k$, then, with probability at least $1 - \delta$, there exists $k' \in (k, k+T_k]$ such that*
$$\bar{f}(\hat{\mathbf{x}}_{k'}) \geq \max_{\mathbf{x} \in \bar{R}_{\bar{L}_{\mathrm{t}}+\epsilon}(S_0)} \bar{f}(\mathbf{x}) - \epsilon.$$

*Proof.*

Let $\mathbf{x}_{k'}^* := \arg\max_{\mathbf{x} \in S_{k'}} \bar{f}(\mathbf{x})$. Note that $\mathbf{x}_{k'}^* \in M_{k'}$, since

$$u_{k'}(\mathbf{x}_{k'}^*, 0) \overset{(a)}{\geq} \bar{f}(\mathbf{x}_{k'}^*)$$
$$\geq \bar{f}(\hat{\mathbf{x}}_{k'})$$
$$\overset{(b)}{\geq} l_{k'}(\hat{\mathbf{x}}_{k'}, 0)$$
$$\overset{(c)}{\geq} \max_{\mathbf{x} \in S_{k'}} l_{k'}(\mathbf{x}, 0)$$

(a) and (b): Lemma B.1,

(c): Definition of $\hat{\mathbf{x}}_k$ (11).

We will first show that $\exists k' \in (k, k + T_k], s.t. \bar{f}(\hat{\mathbf{x}}_{k'}) \geq \bar{f}(\mathbf{x}_{k'}^*) - \epsilon$. Assume, to the contrary, that

$$\forall k' \in (k, k + T_k], \bar{f}(\hat{\mathbf{x}}_{k'}) < \bar{f}(\mathbf{x}_{k'}^*) - \epsilon$$

Then, we have, $\exists k' \in (k, k + T_k]$

$$l_{t'}(\mathbf{x}_{k'}^*, 0)$$
$$\overset{(d)}{\leq} l_{k'}(\hat{\mathbf{x}}_{k'}, 0)$$
$$\overset{(e)}{\leq} \bar{f}(\hat{\mathbf{x}}_{k'})$$
$$< \bar{f}(\mathbf{x}_{k'}^*) - \epsilon$$
$$\overset{(f)}{\leq} l_{k'}(\mathbf{x}_{k'}^*, 0),$$

which is a contradiction.

(d): Definition of $\hat{\mathbf{x}}_k$ (11),

(e): Lemma B.1,

(f): Corollary C.2, and $\mathbf{x}_{k'}^* \in M_{k'}$

Finally, $\bar{R}_{\bar{L}_t + \epsilon}(S_0) \subseteq \underline{S}_{k'} \subseteq S_{k'}$, by Lemma C.6 and Lemma C.4 (ii). Therefore, $\exists k' \in (k, k + T_k]$ such that

$$\max_{\mathbf{x} \in \bar{R}_{\bar{L}_t + \epsilon}(S_0)} \bar{f}(\mathbf{x}) - \epsilon \leq \max_{\mathbf{x} \in S_{k'}} \bar{f}(\mathbf{x}) - \epsilon$$
$$= \bar{f}(\mathbf{x}_{k'}^*) - \epsilon$$
$$\leq \bar{f}(\hat{\mathbf{x}}_{k'})$$

$\square$

## C.3 Near-Optimality Proof

*Proof of Theorem 2.6.* Theorem 2.6 is a direct consequence of Corollary C.2, Lemma C.8, and Lemma C.9. $\square$

# D Practical Considerations

## D.1 Trade-off between Safety and Optimality

In this work, we focus on safety critical systems where satisfying the safety constraints has highest priority over finding the optima. Through pessimistically considering change with time in the decision-making process, TVSAFEOPT emphasizes safety in non-stationary conditions at the inevitable expense of optimality. In practice, such sacrifice on optimality can be alleviated by tighter bound of rate of change.

Besides, in the case where safety can be to some extent comprised, which is beyond the focus of this work, constrained BO and its time-varying extension would be a more suitable method to apply. We refer the readers to [50, 51] for further information.

## D.2 Scalability

Explicit considering time can be viewed roughly as adding dimension by 1, Therefore, TVSAFEOPT achieves time adaptation without adding much computational cost. With the increase in dimensionality of the problem, safety constraints might arise across multiple dimensions, from multiple directions at the price of optimality. As our approach is suitable for safety critical conditions, the focus is put on maintaining safety under change, therefore safety considerations "dictate" the optima. Additionally, in practice, the safety functions are modeled independently with a GP, and thus the computational cost scales linearly with the number of constraints.

