# OpenReview forum: "Safe Time-Varying Optimization based on Gaussian Processes with Spatio-Temporal Kernel"
_NeurIPS.cc/2024/Conference — NeurIPS 2024 poster_

### Official Review · Reviewer_yo4J · 2024-07-08

**Soundness:** 3
**Presentation:** 3
**Contribution:** 3
**Rating:** 6
**Confidence:** 3

**Summary:**

The authors propose a time-varying extension of SAFEOPT to overcome the problems of time-varying rewards under time-varying safety constraints.

Under stationarity conditions, optimality guarantees are provided and the numerical simluation shows a (favorable) comparison to the SAFEOPT.

**Strengths:**

1. The paper is very well written and easy to follow.

2. Based on related work, the problems of time-varying rewards under time-varying safety constraints are an open problem in literature, and his paper addresses that.

3. The paper provides formal safety guarantees for their TVSAFEOPT algorithm.

**Weaknesses:**

1. *Some delineation to related work seems rather vague and requires stronger justification.* An example for TVSBO: the time-variable and temporal aspect of the kernel can just as well be interpreted as context using existing results. Perhaps a table would help here to highlight key aspects.

2. *Lack of real-world data experiments and comparison to related work.* To support the downsides of existing approaches, an empirical comparison to existing TVSBO approaches mentioned in the related work section would be needed.

3. The *empirical results could be more convincing* by adding a variety of initial safe sets and revised plots. The current plots/results are hard to parse.

4. It would be beneficial if the *theoretical/technical challenge of extending safety to the time-varying case were more detailed*. This would streamline the presentation and help in assessing the impact of the contribution.

**Questions:**

1. *In the Appendix, a spatio-temporal SE kernel is introduced. How is this construction different from using an SE-ARD kernel with a composite variable $z = [x^T,t]^T$?* If I am not mistaken, for the SE-ARD kernel there would be no different than defining a single kernel with $z$.

2. It is mentioned that the Lipschitz constants are to be known beforehand. However, while commonly assumed, *how do you get a hold of an RKHS norm bound $B$ (related to Assumption 2.1) to compute the UCB?*

3. *Could you provide Figure 1 sooner in the manuscript?* It would be super helpful to see this central illustration already on page 2.

**Limitations:**

1. Practicality of the safety guarantee: requiring many Lipschitz constants for both for space and time while also requiring an RKHS norm bound.

2. Theoretical and empirical impact: Lack of comparisons to TVSBO approaches makes the implications of the contribution unclear both theoretically and empirically.

---

> ### Author Rebuttal · Authors · 2024-08-07
>
> **W1**
>
> We thank the reviewer for the insight. We refer the reviewer to Table 1 in the pdf.  The main difference between our method and context-based methods lies in how time is handled in the safe sets. Context-based methods require a safe seed to be available for every context or at every iteration, which may be impractical, as time is not a decision variable. Conversely, out TVSafeOPT ensures safety by tightening the definition of the safe set, and thus even just an initial safe seed would be sufficient for the algorithm to find a safe optimum.
>
>
>
>    **W2**
>
>
>  We provide a comparison of the performance of our proposed algorithm, TVSafeOPT, with the solution of an approximate optimization problem used in practice (baseline) and SafeOPT, in the PDF. Contextual approaches with time as a context would require a safe seed for every context, which is unavailable for the unknown changes in a compressor. Similarly, detecting changes due to degradation in turbomachinery is notoriously difficult [10]. The performance of event-triggered SafeOPT relies on the chosen event detection method, thus making the comparison of performance in the compressor case study inaccurate.
>
> The results of the comparison are shown in Fig. 3 in the attached PDF.
>
>
>   **W3**
>
> We added a comparison of safe sets computed by TVSafeOPT (top row in Fig. 1) and SafeOPT (bottom row in Fig. 1). Because TVSafeOPT takes the possible changes in time into consideration, the safe sets computed by TVSafeOPT are contained in the ground truth safe regions while those computed by SafeOPT have multiple violations. .
>
>
>
> **W4**
>
> Technically, the challenge arises in the non-stationary case from the fact that a safe/unsafe action at one moment might become unsafe/safe in the future. This means that a TVSBO algorithm should be aware of such information in order to find a sub-optimal solution while guaranteeing safety. ETSafeOPT [13] hopes to detect the change with an event trigger.  However, in general, when the reward or safety functions change continuously or frequently, ETSafeOPT suffers from poor performance and even loses empirical safety guarantee.  In contrast, context-based methods [1,3] well handle this issue with prior knowledge on the time change infused through the temporal kernel. However, existing context-based methods usually require an initial guess of safe set for each context value, which is often unrealistic. Our algorithm does not rely on change detection and overcomes the challenge related to the safe set initialization at every iteration. This is done by propagating the initial safe set to the future utilizing the temporal kernel.
> Theoretically, proving the safety and near-optimality guarantees for TVSBO algorithms is challenging even in the stationary case because the common proof techniques rely on the monotonicity of the lower bounds, upper bounds, and thus the safe sets, maintained by the algorithm. We lose such monotonicity by design of the algorithm. Instead we manage to circumvent this issue by proposing an lower bound of the safe set which is non-shrinking and converging. Furthermore, near-optimality guarantee in the non-stationary case is a very promising yet challenging problem. The first open-end question to answer is what kind of convergence property under what conditions we can prove for the non-stationary case.
>
>
>  **Q1**
>
> Indeed, there is no technical difference in the two notations, we kept $x$ and $t$ separate to emphasise that $t$ indicates time and, as such, it is not a decision variable.
>
> **Q2**
>
>  Indeed, the Lipschitz constant is usually unknown in realistic optimization scenarios. Its value can be obtained based on information about the capacity of the chosen kernel [7]. Such information, however,  may be unavailable, making the choice a challenge. Oftentimes, however, an educated guess allows overcoming the aforementioned challenge. The authors in [12] have shown that in practice, it is often sufficient to choose an upper bound on $B$ [7], $\beta\geq 2$. This is needed to approximate the Lipschitz constant by an optimistic upper bound. Given that our method relies on Gaussian processes, we made a localized approximation of what has been proposed in [8].
>
> **Q3**
>
> Thanks. We  will apply this change.
>
>
> **L1**
>
> We refer the reviewer to Q2.
>
>  **L2**
>
>  To make our contribution clear, we have now added an overview of the key aspects of existing methods for time-varying safe BO in Table 1 in the PDF. See also Fig, 2,3
>
>
> [1] F. Berkenkamp, A. Krause, A. P. Schoellig, \emph{Bayesian optimization with safety constraints: safe and automatic parameter tuning in robotics},Machine Learning 2021;
>
> [2] A. Holzapfel, P. Brunzema, S. Trimpe, \emph{Event-triggered safe Bayesian optimization on quadcopters}, L4DC 2024;
>
> [3] C. K\"onig, M. Turchetta, J. Lygeros, A. Rupenyan, and A. Krause, Safe and efficient model-free adaptive control via Bayesian optimization, ICRA, 2021;
>
> [4] C. K\"onig, M. Ozols, A. Makarova, E. C. Balta, A. Krause, and A. Rupenyan, {Safe risk-averse Bayesian optimization for controller tuning, 2023;
>
> [5] D. Widmer, D. Kang, B. Sukhija, J. H\"ubotter, A. Krause, and S. Coros. Tuning legged locomotion controllers via safe Bayesian optimiztion, CoRL, 2023.
>
> [7] J. Bergstra, R. Bardenet, Y. Bengio, B. Kegl, Algorithms for hyper-parameter optimization, NIPS, 2021
>
> [8] F. Berkenkamp,  A. P. Schoellig, A. Krause, Safe controller optimization for quadrotors with Gaussian processes, ICRA, 2016.
>
>
> [10] Y. Li, P. Nikitsaranont, Gas turbine performance prognostic for condition-based manteinance, 2009.
>
> [12] C. K\"onig, M. Turchetta, J. Lygeros, A. Rupenyan, A. Krause, Safe and efficient model-free adaptive control via Bayesian optimization, ICRA, 2021
>
> [13] A. Holzapfel, P. Brunzema, S. Trimpe, Event-triggered safe Bayesian optimization on quadrotors, L4DC, 2024
>
> [14] M. Guptca, R. Wadhvani, A Rasool. Comprehensive analysis of change-point dynamics detection in time series data: A review,  2024

---

> > ### Comment · Reviewer_yo4J · 2024-08-11
> > **Reviewer Reply for Submission20789 by Reviewer yo4J**
> >
> > I thank the authors for the rebuttal and the answers to my questions and concerns, and also appreciate them including a table for related work.
> >
> > Since I was already on the positive side in my initial review, so I prefer to keep my overall score with an increases in parts of evaluation.

---

### Official Review · Reviewer_g7A9 · 2024-07-12

**Soundness:** 3
**Presentation:** 3
**Contribution:** 3
**Rating:** 5
**Confidence:** 3

**Summary:**

This paper presents a safe Bayesian optimization algorithm TVSAFEOPT with a spatial-temporal kernel and time Lipschitz constants, which improves on SAFEOPT with time-varying reward and safety constraints. The optimality guarantee is proved for the stationary case and the safety guarantee for more general settings. The method is tested on a synthetic problem and gas compressors.

**Strengths:**

1. The use of a spatio-temporal kernel in Bayesian optimization for time-varying safety constraints is novel.
2. A formal proof of safety and optimality guarantee under certain assumptions.

**Weaknesses:**

1. More discussion on how to make a tradeoff between optimality and safety is encouraged.
2. Will this conservatism in safety become too large in high-dimensional problems?
2. The method to choose the proper initial safe set and kernel parameters is unclear.

**Questions:**

1. How to find an initial safe set for complex problems?
2. How to find the kernel parameter for each task?
2. What is the computational complexity compared to other BO baselines?

**Limitations:**

The societal impact is discussed.

---

> ### Author Rebuttal · Authors · 2024-08-07
>
> We thank the reviewer for the useful feedback, and the positive assessment of the paper. We provide a point-by-point answers to all raised suggestions, comments, and questions.
>
>  **W1**
>
> We thank the reviewer for this suggestion. In the paper we are focused on safety critical systems where satisfying the safety constraints has highest priority over finding the optimum. The focus on safety is illustrated in Fig. 1 in the paper, where the proposed algorithm follows the safe set, even though the safe set changes over time. In particular, at time $t=170$, the algorithms takes decisions within a safe set that is smaller than the initial safe set, further emphasizing the safety at the expense of optimality. We will discuss these aspects in the Appendix C of the revised paper.
>
> **W2**
>
> We thank the reviewer for raising this question. As our approach is suitable for safety critical conditions, the focus is put on maintaining safety under change, therefore safety considerations ``dictate'' the optima. If the dimensionality of the problem increases, safety constraints might arise across multiple dimensions, from multiple directions at the price of optimality. This clarification will be included in Appendix C.
>
>
>    **W3**
>
>  Indeed, the choice of the safe set and the kernel is part of ongoing research [6]  and typically requires some extended knowledge about the underlying functions[7.8]. A common choice for the kernel is a squared exponential kernel that intuitively puts emphasis on the points that are close to each other. The meaning of ``close'' is further defined by hyperparameters, which can be obtained for instance through maximum likelihood estimation [9].
>
>  Regarding the choice of the safe set, often a single safe point is already sufficient [Ch. 16, 10]. One of our assumptions is that at least one safe point is available in the initial safe set, and this is most often the case in practice.  In most application it is reasonable to assume some prior knowledge is available, which in turn can be exploited in the initialization phase. This assumption is a direct consequence of the underlying principle that both the objective and the constraint functions can be measured. For example, in the compressor optimization case, the safe point is obtained from domain knowledge - an equal distribution of the gas is a safe point, but may be far from optimal if the compressors are dissimilar [10].
>
> We further point out that most of the algorithms in this domain  require the safe set to be reinitialized after each iteration (see Table 1 in the PDF and the associated references at the end of this paragraph). In contrast, TVSafeOPT requires only one initial safe set at $t=0$. As indicated, we will dedicate a subsection in Appendix C to address these points.
>
>
>
>   **Q1**
>
>  In complex setting it is still reasonable to assume that at least one single safe point is available. As we also mention in the answer to Weakness 3 raised from the reviewer, this would be already sufficient for TVSafeOPT to run and to guarantee optimality within the safe region that can either  enlarge or shrink along time (see Fig. 1 in the submitted manuscript).
>
>  For example, in the compressor optimization case, the safe point is obtained from domain knowledge - an equal distribution of the gas is a safe point, but may be far from optimal if the compressors are dissimilar [11].
>
>
> **Q2**
>
> The choice of the kernel and the hyperparameters of the algorithm is a part of ongoing research [6]  and typically requires some extended knowledge about the underlying functions [7,8]. A common choice for the kernel is a squared exponential kernel that intuitively puts emphasis on the points that are close to each other. The meaning of ``close'' is further defined by hyperparameters, which can be obtained for instance through maximum likelihood estimation [10]. One way to adjust the hyperparameters is to collect data in advance and optimize the hyperparameters via maximum likelihood optimization, following standard GP regression methods.
>
> **Q3**
>
> Through observations during the simulations, the computational complexity of each iteration of TVSafeOpt is only slightly heavier than SafeOpt; The sample complexity of the algorithm is similar to SafeOpt [1] in the stationary case and scales as $\mathcal{O}(\varepsilon^{-2})$, where $\varepsilon$ is the accuracy threshold for the optimization. In the non-stationary case, it is a still open problem to bound the sample complexity.
>
>
>
>
>
> **References**
>
> [1] F. Berkenkamp, A. Krause, A. P. Schoellig, \emph{Bayesian optimization with safety constraints: safe and automatic parameter tuning in robotics},Machine Learning 2021;
>
> [2] A. Holzapfel, P. Brunzema, S. Trimpe, \emph{Event-triggered safe Bayesian optimization on quadcopters}, L4DC 2024;
>
> [3] C. K\"onig, M. Turchetta, J. Lygeros, A. Rupenyan, and A. Krause, \emph{Safe and efficient model-free adaptive control via Bayesian optimization}, ICRA, 2021;
>
> [4] C. K\"onig, M. Ozols, A. Makarova, E. C. Balta, A. Krause, and A. Rupenyan, \emph{Safe risk-averse Bayesian optimization for controller tuning}, IEEE Rob. and Autom. Letters, 2023;
>
> [5] D. Widmer, D. Kang, B. Sukhija, J. H\"ubotter, A. Krause, and S. Coros. Tuning legged locomotion controllers via safe Bayesian optimiztion, CoRL, 2023.
>
> [6] C. Fiedler, J. Menn, L. Kreisk ̈other, S. Trimpe, On safety in safe Bayesian Optimization, arXiv, 2024
>
>
> [7] J. Bergstra, R. Bardenet, Y. Bengio, B. Kegl, Algorithms for hyper-parameter optimization, NIPS, 2021
>
> [8] F. Berkenkamp,  A. P. Schoellig, A. Krause, Safe controller optimization for quadrotors with Gaussian processes, ICRA, 2016.
>
> [9] Y. Sui, A. Gotovos, J. Burdick, A. Krause, Safe exploration for optimization with Gaussian processes, PMLR, 2015.
>
> [10] Y. Li, P. Nikitsaranont, Gas turbine performance prognostic for condition-based manteinance. Applied Energy, 2009.
>
> [11] B. G. Liptak, Instrument Engineers' Handbook, vol. 2, 2005

---

> > ### Comment · Reviewer_g7A9 · 2024-08-11
> >
> > Thank you for the rebuttal. Based on the responses I am comfortable with my current score for this paper.

---

### Official Review · Reviewer_o5DK · 2024-07-13

**Soundness:** 2
**Presentation:** 2
**Contribution:** 2
**Rating:** 5
**Confidence:** 2

**Summary:**

The paper introduces the TVSAFEOPT algorithm, which is based on Gaussian processes with spatio-temporal kernels, designed specifically for optimizing time-varying rewards under time-varying safety constraints. The algorithm provides formal safety guarantees in a general time-varying setting, ensuring safety even when exploring non-stationary safe regions. It robustly subtracts safety margins to prevent unsafe decisions, adapting in real-time to changing environments. Furthermore, they provide optimality guarantees for locally stationary optimization problems, ensuring near-optimal solutions when the optimization problem becomes stationary.

**Strengths:**

They provide formal safety guarantees in dynamic environments, ensuring safe decision-making even in non-stationary settings.

Additionally, the algorithm offers optimality guarantees for stationary optimization problems, enhancing its reliability and performance

Extensive numerical simulations were provided to validate the proposed approach.

**Weaknesses:**

They extend the Safeopt algorithm from literature. However, it is clear on what are the additional contributions and difference between these two different approaches.

**Questions:**

-

**Limitations:**

-

---

> ### Author Rebuttal · Authors · 2024-08-07
>
> **W1:They extend the Safeopt algorithm from literature. However, it is clear on what are the additional contributions and difference between these two different approaches.**
>
> We thank the reviewer for the positive assessment of our paper, and for their constructive feedback. We now provide a table (see Table 1 in the PDF and refer to References) to give a clear overview of the contributions of our proposed approach TVSafeOPT, compared  with other methods associated with time-varying safe optimization. In contrast to safe learning methods based on contextual Bayesian optimization, such as Contextual SafeOPT [1], which rely on using time as a context and thus require a safe seed for every context, our TVSafeOPT requires only an initial safe seed. Adaptive Goal Oriented Safe Exploration (A-GoOSE) can also handle a single initial seed, but without providing theoretical guarantees on safety across time [3,4]. Furthermore, compared to event-triggered methods such as ETSafeOPT [2], our method requires the initial safe set to be provided only once, at the initial time instant, $t=0$. In contrast, ETSafeOPT requires a new safe state initialisation after each iteration. Moreover, we are able to provide  both safety and convergence guarantees to the optimal solution within the safe set, while ETSafeOPT guarantees only safety.
>
> **References**
>
>
> [1] F. Berkenkamp, A. Krause, A. P. Schoellig, \emph{Bayesian optimization with safety constraints: safe and automatic parameter tuning in robotics},Machine Learning 2021;
>
> [2] A. Holzapfel, P. Brunzema, S. Trimpe, \emph{Event-triggered safe Bayesian optimization on quadcopters}, L4DC 2024;
>
> [3] C. K\"onig, M. Turchetta, J. Lygeros, A. Rupenyan, and A. Krause, \emph{Safe and efficient model-free adaptive control via Bayesian optimization}, ICRA, 2021;
>
> [4] C. K\"onig, M. Ozols, A. Makarova, E. C. Balta, A. Krause, and A. Rupenyan, \emph{Safe risk-averse Bayesian optimization for controller tuning}, IEEE Rob. and Autom. Letters, 2023;
>
> [5] D. Widmer, D. Kang, B. Sukhija, J. H\"ubotter, A. Krause, and S. Coros. Tuning legged locomotion controllers via safe Bayesian optimiztion, CoRL, 2023.

---

> > ### Comment · Reviewer_o5DK · 2024-08-13
> > **Response to authors**
> >
> > Thanks for the clarifications. I would like to maintain my score

---

### Author Rebuttal · Authors · 2024-08-07

Dear Chairs,

Dear Reviewers,


Thank you for the thoughtful feedback on our manuscript. All three reviewers found our results of interest to the wide readership of NeurIPS. In particular, the reviewers appreciated the theoretical guarantees for safety and optimality of our proposed TVSafeOPT algorithm.

The main reservation of the reviewers concerns the choice of the initial safe set and robustness of our algorithm with respect to  the initial safe set, together with a more clear explanation on how the proposed algorithm differs from the current time-varying safe learning algorithms with direct comparison. Other suggestions to further improve the manuscript ask for a clarification of the technical challenges of the time varying setting and a discussion on trading off safety and optimality.


We have now revised the manuscript to address all the comments from the reviewers, in particular:

1) We show via additional experiments how our algorithm is robust with respect to perturbation on the initial safe set;

2) We make it clear how our algorithm departs from the other time-varying optimization algorithms in the literature concerning optimization in safety critical settings, and provided Table 1 to make an overview about the state of the art and the contribution of our paper.
    We also show, experimentally, the contribution of our algorithm with respect the the baseline SafeOPT and to a time-varying BO baseline, event-triggered BO;

3) We describe the technical challenges in providing theoretical guarantees for the case in which the objective function does not reach a steady state.

We hope that these revisions, as well as the individual answers provided to all comments and questions improve the presentation of our algorithm. We appreciate the opportunity to resubmit our manuscript for potential publication in NeurIPS and thank you in advance for your time.


Yours sincerely,

The authors

---

### Author Response · Authors · 2024-08-07

Dear Program chairs,

It looks like our uploaded pdf and general answer is not visible to the reviewers. Could this be the case, and is it possible to make it visible?
I also saw this in other rebuttals which I am reviewing where there is a reference to a pdf but no pdf uploaded.

Thank you for your help.

---

### Decision · Program_Chairs · 2024-09-25

**Decision:**

Accept (poster)

**Comment:**

This paper extends the SafeOpt algorithm for online safe optimization when the safety-critical system is time-varying. The authors provide theoretical analysis on safe optimization with a spatio-temporal kernel, together with synthetic experiments and a realistic case study with gas compressors.

The meta-reviewer agrees with all reviewers that the paper has reached the bar for NeurIPS. The AC would recommend the authors to improve the paper towards a camera-ready version based on the rebuttals and discussions.